# Cortical neuroprosthesis-mediated functional ipsilateral control of locomotion in rats with spinal cord hemisection

Elena Massai[1†], Marco Bonizzato[1,2†], Isley De Jesus[1,2‡], Roxanne Drainville[1,2‡], Marina Martinez[1,2]*

[1]Département de Neurosciences, Groupe de recherche sur la Signalisation Neurale etla Circuiterie (SNC) and Centre Interdisciplinaire de Recherche sur le Cerveau etl'Apprentissage (CIRCA), Université de Montréal, Montréal, Canada; [2]CIUSSS du Nord-de-l'Île-de-Montréal, Montréal, Canada

## eLife Assessment

The contributions of ipsilateral cortical pathways to motor control are yet not fully understood. Here, the authors present **important** insights into their role in locomotion following unilateral spinal cord injury. Their data provide **convincing** evidence in rats that stimulation of ipsilateral motor cortex improves the injured side's ability to support weight and leads to improved locomotion, a result that may inspire new treatments for spinal or cerebral injuries.

*For correspondence: marina.martinez@umontreal.ca

[†]These authors contributed equally to this work
[‡]These authors also contributed equally to this work

**Abstract** Control of voluntary limb movement is predominantly attributed to the contralateral motor cortex. However, increasing evidence suggests the involvement of ipsilateral cortical networks in this process, especially in motor tasks requiring bilateral coordination, such as locomotion. In this study, we combined a unilateral thoracic spinal cord injury (SCI) with a cortical neuroprosthetic approach to investigate the functional role of the ipsilateral motor cortex in rat movement through spared contralesional pathways. Our findings reveal that in all SCI rats, stimulation of the ipsilesional motor cortex promoted a bilateral synergy. This synergy involved the elevation of the contralateral foot along with ipsilateral hindlimb extension. Additionally, in two out of seven animals, stimulation of a sub-region of the hindlimb motor cortex modulated ipsilateral hindlimb flexion. Importantly, ipsilateral cortical stimulation delivered after SCI immediately alleviated multiple locomotor and postural deficits, and this effect persisted after ablation of the homologous motor cortex. These results provide strong evidence of a causal link between cortical activation and precise ipsilateral control of hindlimb movement. This study has significant implications for the development of future neuroprosthetic technology and our understanding of motor control in the context of SCI.

## Introduction

Cortical commands primarily regulate contralateral limb movements. This lateralization bias is reflected (1) anatomically, by a majority of crossed corticospinal tract (CST) projections (*Hicks and D'Amato, 1975*), (2) electrophysiologically, by a predominance of contralateral muscle recruitments by cortical stimulation (*Kwan et al., 1978*), (3) functionally, by contralateral deficits induced by cortical lesions (*Passingham et al., 1983*). However, lateralization of cortical control is incomplete, yet there is limited evidence on the functional significance of cortical ipsilateral regulation of movement (*Montgomery*

*et al., 2013*). Ipsilateral impairments have been reported after unilateral cortical injury or transient interference (e.g. via transcranial magnetic stimulation) accompanied with increased cortical activity from the opposite hemisphere (*Blasi et al., 2002*; *Chen et al., 1997b*; *Johansen-Berg et al., 2002*; *Jones et al., 1989*; *Kim et al., 2003*; *Marque et al., 1997*; *Yarosh et al., 2004*). Nevertheless, the role of the motor cortex in regaining control over ipsilateral movements after injury remains controversial (*Caramia et al., 2000*; *Chen et al., 1997a*; *Dancause et al., 2006*; *Hallett, 2001*; *Hummel and Cohen, 2006*; *Jankowska and Edgley, 2006*; *Serrien et al., 2004*; *Stoeckel and Binkofski, 2010*; *Turton et al., 1996*). While the motor cortex primarily controls movements on the opposite side of the body (contralateral), it also contains a representation of both unilateral and bilateral movements in the same-side cortical hemisphere (ipsilateral) (*Aizawa et al., 1990*; *Ames and Churchland, 2019*; *Bundy et al., 2018*; *Cisek et al., 2003*; *Diedrichsen et al., 2013*; *Donchin et al., 1998*; *Ganguly et al., 2009*; *Ghacibeh et al., 2007*; *Heming et al., 2019*; *Kawashima et al., 1994*; *Merrick et al., 2022*; *Tinazzi and Zanette, 1998*; *Wisneski et al., 2008*). Imaging studies have shown that lower extremities movements and walking, which require efficient bilateral coordination, are associated with bilateral activity in primary sensorimotor cortices and supplementary motor areas (*Miyai et al., 2001*). Yet cortical dynamics underlying locomotion have been primarily studied in relation to contralateral kinematics (*Barroso et al., 2019*; *Bonizzato et al., 2018*; *Brown and Martinez, 2021*; *DiGiovanna et al., 2016*; *Song et al., 2009*; *Yin et al., 2014*). The relationship between cortical commands and locomotion has received attention in the last decades (*Amboni et al., 2013*). In recent studies, we have shown that, after a unilateral spinal cord injury (SCI) in rats (*Bonizzato and Martinez, 2021*) and large spinal contusion injuries in cats (*Duguay et al., 2023*), microstimulation delivered to the contralesional motor cortex in phase coherence with locomotion immediately alleviated contralateral hindlimb deficits. Other studies have shown that not only the cortex proactively controls high-level and goal-oriented motor planning but it is also involved during stereotyped locomotion (*Artoni et al., 2017*; *Bretzner and Drew, 2005*; *Song and Giszter, 2011*; *Song et al., 2009*). Nevertheless, there is still limited evidence showing that cortical networks can be interfaced to probe and augment control of hindlimb movements, especially with respect to ipsilateral cortical contribution.

To address this knowledge gap, we designed a behavioral neuromodulation framework to assess the gait-phase-specific effects of intracortical neurostimulation on ipsilateral hindlimb kinematics during locomotion. We evaluated the immediate modulation of hindlimb trajectory and posture both in intact rats and after a unilateral hemisection SCI. This side-specific lesion preserves most crossed projections from the ipsilateral motor cortex while maximizing the loss of direct efferences from the contralateral motor cortex. In most cases, this lesion also disrupts all spinal tracts descending on the same side as the cortex under investigation at the thoracic level, meaning that the transmission of cortical commands to the ipsilesional hindlimb must depend on crossed descending tracts. As early as 1 week after injury, different modalities of ipsilateral cortical neuroprosthetic stimulation immediately alleviated SCI-induced deficits, including lack of hindlimb support, weak hindlimb extension and flexion, and dragging.

Our functional causal approach to ipsilateral movement directly challenges the classical view whereby ipsilateral motor cortex control of movement is epiphenomenal and functionally limited. We demonstrate that the ipsilateral motor cortex has functional control of hindlimb motor synergies and that its action can reverse SCI locomotor deficits, independently from the homologous motor cortex. We then sought to provide a parallel description of the time-course of ipsilateral corticospinal transmission and spontaneous recovery of locomotor function after SCI. We longitudinally acquired and scored hindlimb movements evoked by intracortical stimulation, obtaining chronic ipsilateral 'motor maps' in awake rats. Finally, by unilaterally delivering longer cortical stimulation trains, we show activation of bilateral flexion-extension rhythms, and that this control property is transiently lost and then recovered in our SCI model.

## Results

In this study, we developed cortical neurostimulation protocols to investigate the role of the motor cortex in controlling ipsilateral hindlimb movements. Our primary objectives were to determine whether this stimulation could modulate ongoing locomotor patterns in intact conditions and immediately alleviate motor deficits following hemiparesis induced by a lateralized SCI. To achieve this, we applied intracortical stimulation in synchrony with specific gait phases, precisely timed to coincide

with either the contralateral or ipsilateral foot lift. We used real-time processing of muscle activity to predict the timing of these gait events, as in our previous work (*Bonizzato and Martinez, 2021*). Our experimental model involved inducing selective unilateral hindlimb deficits through a thoracic lateral hemisection of the spinal cord (*Brown and Martinez, 2019*). This injury results in transient paralysis of the hindlimb on the side of the spinal lesion due to the loss of major supraspinal inputs. Importantly, the ipsilesional motor cortex, which corresponds to the left motor cortex and left leg in our study, retains most of its crossed connections to the sublesional spinal circuits.

For clarity, throughout the manuscript, we will use the terms 'ipsilateral' and 'ipsilesional' to refer to the left implanted motor cortex and the left leg, both located on the same side of the spinal hemisection. Conversely 'contralateral' and 'contralesional' will exclusively pertain to the right motor cortex and right leg. In brief, left = ipsi-; right = contra-.

## Phase-coherent stimulation in intact rats

Online detection of muscle activation was used to predict gait events and consequently trigger short-train intracortical microstimulation (ICMS) (40 ms, 330 Hz) through a 32-channel intracortical array implanted into the left motor cortex. The specific channel within the array was chosen based on its ability to generate the strongest contralateral ankle flexion. Stimulation through these channels produced a strong whole-leg flexion movement, with an evident distal component. From visual inspection, all responding electrodes in the array produced contralateral leg flexion, although with different strength of contraction for a fixed stimulation intensity (100 µA). Moreover, some sites did not present a distal movement component, failing in eliciting ankle flexion and resulting in a generally weaker proximal flexion.

To assess the effects of ICMS on leg trajectory and locomotor behavior, we conducted experiments with six intact rats (*Figure 1A*). Our findings demonstrated that the modulation of gait was dependent on the timing of the stimulation. The most significant effects were observed when stimuli were delivered during the preparation and early execution of the right swing phase of gait. These effects included an increase in right hindlimb flexion (*Figure 1B*).

Furthermore, when we synchronized the stimulation with the timing of the right foot lift (within ±75 ms, referred to as 'phase coherence', as previously described in *Bonizzato and Martinez, 2021*), we observed specific alterations in gait patterns. These alterations included an increase in right step height (+119 ± 37% of the spontaneous level, p=4E-4, t-test, phase-coherent vs. off-timing). Additionally, the gait pattern modifications resulted in a contralateral swing dominance (i.e. significantly longer than that of the opposite limb by +22 ± 6%, p=0.03, t-test) and an ipsilateral stance dominance (+10 ± 3%, p=0.01, t-test) (*Figure 1B*).

In our investigation of the effects of modulating phase-coherent stimulation amplitudes, we observed that there were no discernible effects on posture (*Figure 1C*) or ipsilateral kinematics (*Figure 1D*) across all intact rats. However, the stimulation did have a notable impact on various aspects of contralateral limb movement during walking. Specifically, as we increased the stimulation amplitudes, we observed linear increases in several parameters related to contralateral hindlimb movement. These parameters included right step height (+157 ± 13%, p=4E-5, t-test), flexion velocity (+107 ± 21%, p=2E-4, t-test), and swing (30 ± 3%, p=5E-4, t-test) and stance (14 ± 2%, p=0.0026, t-test) asymmetry indexes. The relationships between stimulation amplitudes and these parameters were characterized by high variance accounted for (VAF) values, ranging from [79±5, 78±6, 80±4, 73±9]% (*Figure 1E and F*).

## Phase-coherent stimulation in SCI rats

We then tested the immediate impact of cortical stimulation in modulating locomotor output in seven rats, each exhibiting various hemisection profiles (*Figure 2A*, blue star) and ladder scores at week 1 (*Figure 2B*).

Following a left spinal hemisection at the thoracic level (*Figure 3A*), rats displayed ipsilateral hindlimb motor deficits corresponding to the same side as the lesion. About 1 week after the injury (5–10 days depending on the injury severity), once the animal had regained alternated hindlimb stepping (*Figure 3B and C*), we assessed treadmill locomotion. The observed deficits included a lack of ipsilateral hindlimb support, as well as weaker ipsilateral flexion and extension, leading to

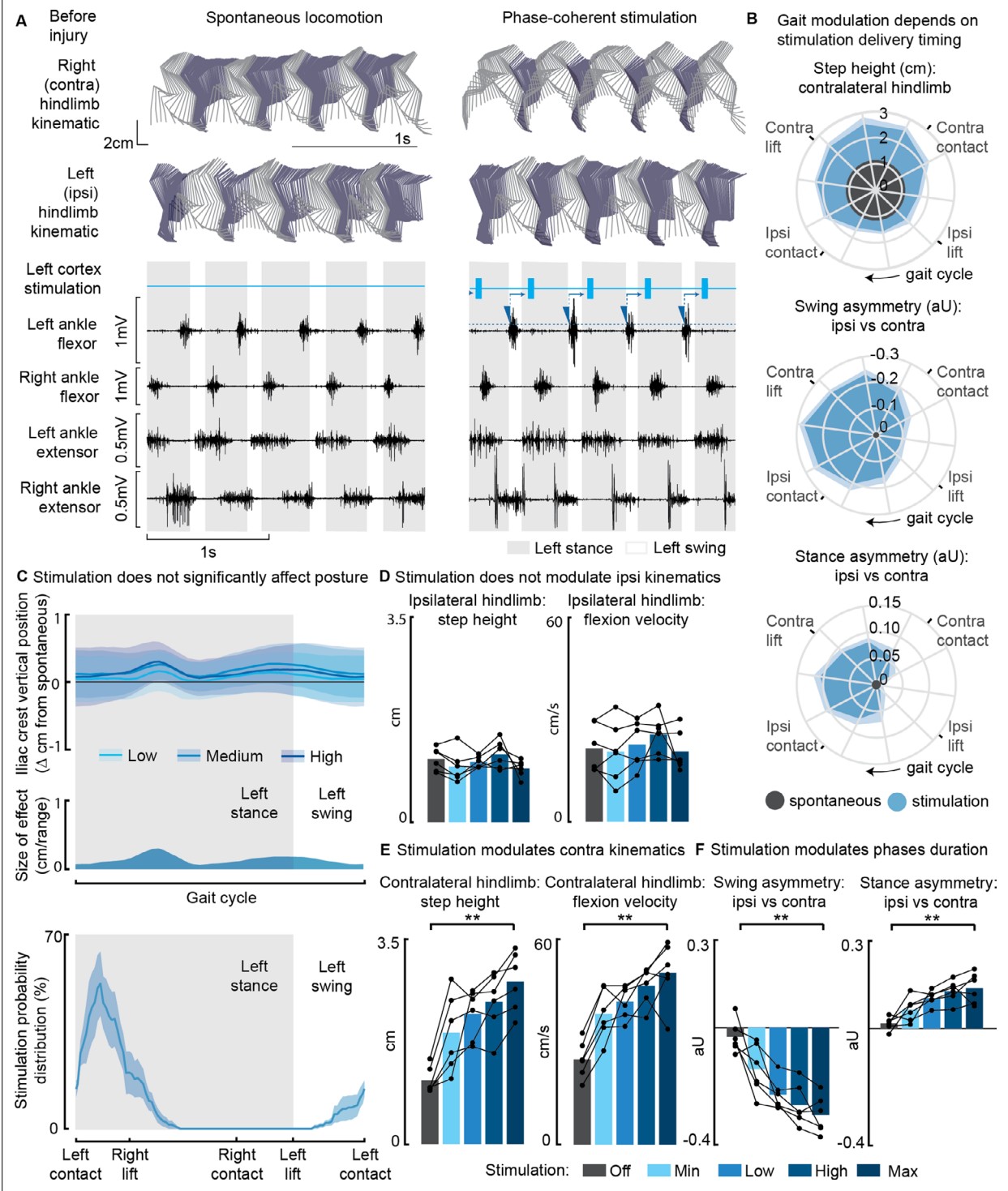

**Figure 1.** Phase-coherent intracortical stimulation modulated contralateral kinematics in intact rats (n=6 rats). (**A**) Stick diagrams and electromyographic (EMG) activity during spontaneous locomotion and phase-coherent stimulation. The stimulation was triggered by ipsilateral ankle flexor activation and was delivered during the late contralateral stance (early ipsilateral stance). On the left ankle trace, the dotted line represents a threshold value for the ankle EMG to cause a trigger, represented by a blue triangle. The stimulation delivery follows a fixed delay indicated with an arrow. (**B**) Polar plots showing contralateral step height in cm and gait phase asymmetries in arbitrary units (aU, calculated as Δ% length) for stimulation delivered with different timings along the whole gait cycle. Positive asymmetry index values refer to ipsilateral-side dominance. For ease of reading, the radial axis of the swing symmetry plot has been inverted (outer values are negative). For the three polar plots, the most effective kinematic neuromodulation corresponds to the largest radius. The gait cycle progresses clockwise. (**C**) Analysis of the effects of cortical stimulation on the posture of rats (top) and

*Figure 1 continued on next page*

*Figure 1 continued*

experimental stimulation distribution (bottom). Posture is shown as the height of the ipsilateral iliac crest during the gait cycle, which was not modulated by increasing cortical stimulation amplitude, indicated as Low, Medium, and High to represent 33%, 66%, and 100% of the functional stimulation range, defined from motor threshold to maximum comfortable stimulus. (**D**) Characterization of ipsilateral kinematics. Ipsilateral step height and flexion speed were not affected by increasing cortical stimulation amplitudes. (**E**) Modulation of contralateral kinematics. Contralateral step height and flexion speed were linearly increased with greater stimulation amplitudes. (**F**) Modulation of bilateral gait phase duration. The absolute values of swing and stance asymmetry indexes were linearly increased with greater stimulation amplitudes. Positive asymmetry index values refer to ipsilateral-side dominance. The data are represented as the mean ± SEM. **$p<0.01$.

The online version of this article includes the following source data for figure 1:

**Source data 1.** Source data for *Figure 1B, D-F*.

asymmetries in the gait pattern (ipsilateral swing dominance 29 ± 4% and right stance dominance 16 ± 3%, *Figure 3E*).

Phase-coherent stimulation of the ipsilateral motor cortex (see scheme in *Figure 3A*) enhanced contralateral step height (*Figure 3E*). This effect was behaviorally expressed as a bilateral synergy, characterized by a contralateral hindlimb flexion and an ipsilateral extension. Consequently, ipsilateral weight-bearing was intensified and prolonged, leading to a reversal of the motor deficits and the restoration of a balanced gait phase distribution between the ipsilateral and contralateral hindlimb (*Figure 3B and C*, *Figure 3—figure supplement 1*, *Figure 3—video 1*). The most significant effects were obtained when stimuli were delivered during the preparation and early execution of the right swing, similar to what was observed in the intact condition (*Figure 3—figure supplement 2A*). These strong effects included an increase in contralateral step height (+94 ± 43%, p=0.001, t-test, phase-coherent vs. off-timing stimuli) as well as a counterbalance between swing (p=2E-5, t-test) and stance durations (p=0.0017, t-test) effectively reversing the asymmetry deficit up to 116 ± 11% and 115 ± 9% respectively, compared to intact walking (*Figure 3D* and *Figure 3—figure supplement 2B*). When delivered during the late right or early ipsilateral stance, stimulation amplitude linearly (VAF = 85 ± 4%) modulated right step height (+172 ± 30%, p=1E-4, t-test) and flexion velocity (+115 ± 22%, p=7E-4, t-test, linear fit VAF = 86 ± 4%). In addition, the swing (deficit reversed up to 123 ± 10%, p=0.0018, t-test, linear fit VAF = 80 ± 7%) and stance (deficit reversed up to 125 ± 10%, p=9E-4, t-test, linear fit VAF = 86 ± 5%) asymmetry indexes proportionally decreased (*Figure 3E*).

We confirmed that the modulation of ipsilateral hindlimb kinematics described in this experiment persisted even 1 month after SCI, with all effects in hindlimb extension and swing/stance asymmetry remaining consistent (*Figure 3—figure supplement 3A–C*). The stimulation currents needed to

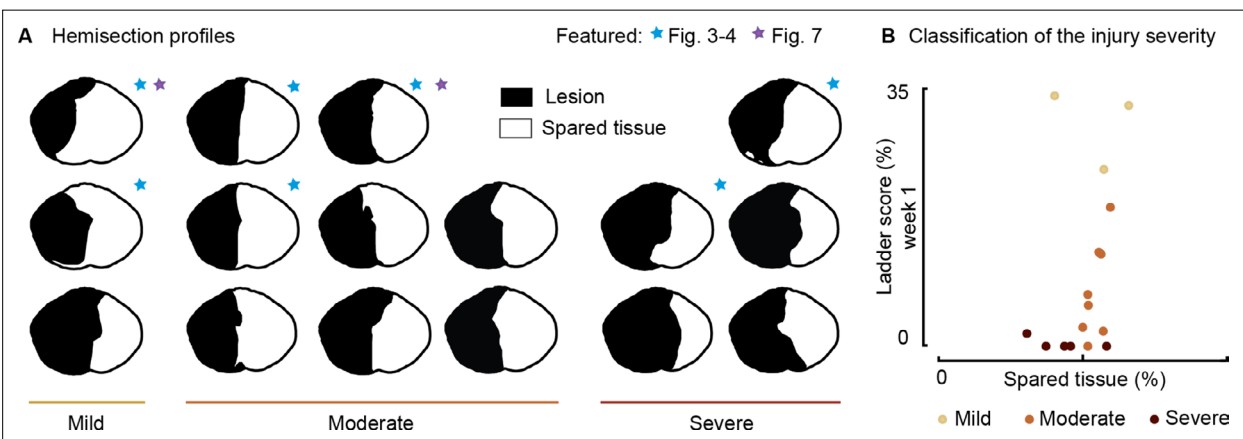

**Figure 2.** Spinal lesion severity (n=16 rats). Related to *Figures 3, 4 and 7*. (**A**) Hemisection profiles at the epicenter level. (**B**) Classification of the injury severity. Injury severity groups were defined according to skilled locomotion performance during ladder crossing 7 days after injury. The injuries were classified as mild: left hindlimb>20% correct paw placement, moderate: left hindlimb<20% correct paw placement and right hindlimb>75% correct paw placement and severe: right hindlimb<75% correct paw placement (bilateral deficits).

The online version of this article includes the following source data for figure 2:

**Source data 1.** Source data for *Figure 2B*.

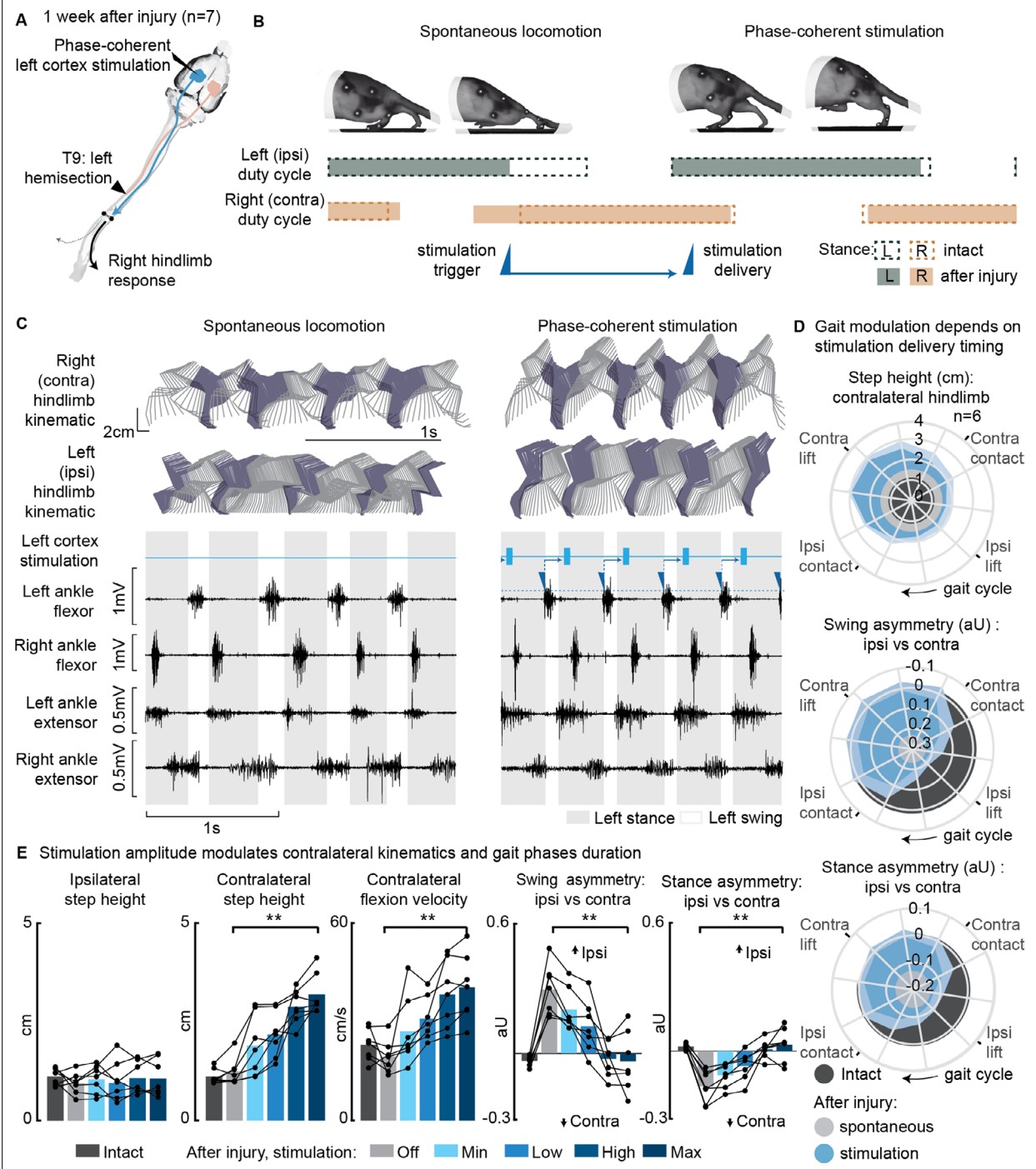

**Figure 3.** Phase-coherent intracortical stimulation alleviated locomotor deficits 1 week after injury (n=7 rats). (**A**) A schematic representation of the injury and neurostimulation model showing the thoracic left hemisection (**T9**) and left (ipsilesional) motor cortex stimulation. (**B**) A schematic representation of spontaneous locomotion and phase-coherent stimulation effects on postural changes, gait phase duration and alternation, as well as stimulation trigger and delivery timings. The stimulation, triggered in correspondence with the ipsilateral lift and delivered just before the contralateral lift, resulted in a stronger contralateral swing and a synchronous stronger ipsilateral stance. (**C**) Stick diagrams and electromyographic (EMG) activity during spontaneous locomotion and phase-coherent stimulation. The stimulation was triggered by ipsilateral ankle flexor activation and was delivered during the late contralateral stance (early ipsilateral stance). (**D**) Polar plots showing contralateral step height (cm) and gait phase asymmetry variations (aU, calculated as Δ% length) for stimulation delivered at different timings along the whole gait cycle. Positive asymmetry index values refer to ipsilateral-side dominance. For ease of reading, the radial axis of the swing symmetry plot has been inverted (outer values are negative). For the three polar plots, the condition of strongest neuromodulation corresponded to the largest radius. Gait phase symmetry, highly affected during spontaneous locomotion,

*Figure 3 continued on next page*

*Figure 3 continued*

was recovered for stimulation delivered after the ipsilateral contact and before the contralateral contact. The gait cycle progresses clockwise. (**E**) The contralateral kinematics and gait phase durations were linearly modulated with increasing stimulation amplitudes. Positive asymmetry index values refer to ipsilateral-side dominance. Phase-coherent stimulation generated an increase in the step height and flexion speed of the contralateral hindlimb and mediated the recovery of the physiological symmetry between the ipsilateral and contralateral swing and stance phases. The data are represented as the mean ± SEM. **p<0.01. The hemisection profiles of the seven rats are identified by a blue star in *Figure 2A*.

The online version of this article includes the following video, source data, and figure supplement(s) for figure 3:

**Source data 1.** Source data for *Figure 3D and E*.

**Figure supplement 1.** Graphical scheme summarizing the results.

**Figure supplement 2.** Aggregate timing characterization of phase-coherent stimulation (n=7 rats).

**Figure supplement 2—source data 1.** Source data for *Figure 3—figure supplement 2B*.

**Figure supplement 3.** Persistent effects of phase-coherent cortical stimulation 1 month after spinal cord injury (SCI) (n=5 rats).

**Figure supplement 3—source data 1.** Source data for *Figure 3—figure supplement 3*.

**Figure 3—video 1.** Cortical neuroprosthesis-mediated control of ipsilateral hindlimb extension.

https://elifesciences.org/articles/92940/figures#fig3video1

achieve this modulation decreased over time (*Figure 3—figure supplement 3D*), while electrode impedances generally remained stable (*Figure 3—figure supplement 3E*).

Next, we examined the effects of phase-coherent stimulation on muscle activity (*Figure 4A*). Following the injury, the ipsilesional ankle extensor muscle exhibited significant alterations during spontaneous locomotion (–72 ± 4% burst duration, p=0.007, t-test, –92 ± 2% total activation, p=0.002, t-test, compared to intact conditions, *Figure 4B*). However, phase-coherent stimulation reinstated the function of this muscle, leading to increased burst duration (recovered 90 ± 18% of the lost burst duration, p=0.004, t-test, *Figure 4B*) and total activation (recovered 56 ± 13% of the total activation, p=0.014, t-test, *Figure 3B*), with the degree of improvement linearly correlated with the applied stimulus amplitude (VAF=[84±7, 84±10]%).

After the injury, rats displayed a low posture due to the loss of weight acceptance on their ipsilateral hindlimb, and the severity of these postural deficits depended on the SCI severity (*Figure 5A*). However, during the recovery process, postural compensation occurred, leading to a notably elevated posture 1 month after SCI (*Figure 5B*). Phase-coherent stimulation, administered in the early ipsilateral stance phase, immediately alleviated postural deficits 1 week after injury, and the iliac crest height increased proportionally with higher stimulation amplitudes (p=0.03, t-test, VAF = 76 ± 9%, *Figure 5C*).

## Awake motor maps

In 12 awake rats allowed to spontaneously recover in their cage, we collected cortical maps, measuring from the ipsilesional motor cortex, which served as for measuring a proxy of corticospinal transmission to both hindlimbs for 8 weeks following SCI (*Figure 6A*).

By visually inspecting the responses elicited by stimulation delivered through each of the array electrodes, we categorized movements as proximal or distal. This classification was based on whether the ankle participated in the evoked response or if the movement was restricted to the proximal hindlimb. Each leg was scored independently.

Initially, the injury led to a substantial decrease in corticospinal transmission on both sides: 5 days after the injury, the ipsilateral (left motor cortex to left hindlimb) and contralateral (left motor cortex to right hindlimb) transmission decreased by approximately –90 ± 7% and –53±13% (p=[2E-4, 5E-4], Wilcoxon signed-rank test, *Figure 6B*), respectively. The size of the contralateral map (i.e. the number of responding sites on the implanted array) substantially increased by 2 weeks (+264 ± 20%) and remained consolidated 8 weeks after injury (+250 ± 21%, p=2E-4, Wilcoxon signed-rank test, *Figure 6B*). Over time, the representation of ipsilateral hindlimb movements significantly increased compared to the intact condition (+115 ± 26%, p=0.002, Wilcoxon signed-rank test, *Figure 6B*). The upregulation of corticospinal transmission and postural changes during spontaneous locomotion predominantly occurred within the same timeframe, specifically 1–2 weeks after SCI (*Figure 6A–D*). Between weeks 1 and 2, 91 ± 22% of the overall postural correction (*Figures 5B and 6C*) and 71 ± 2%

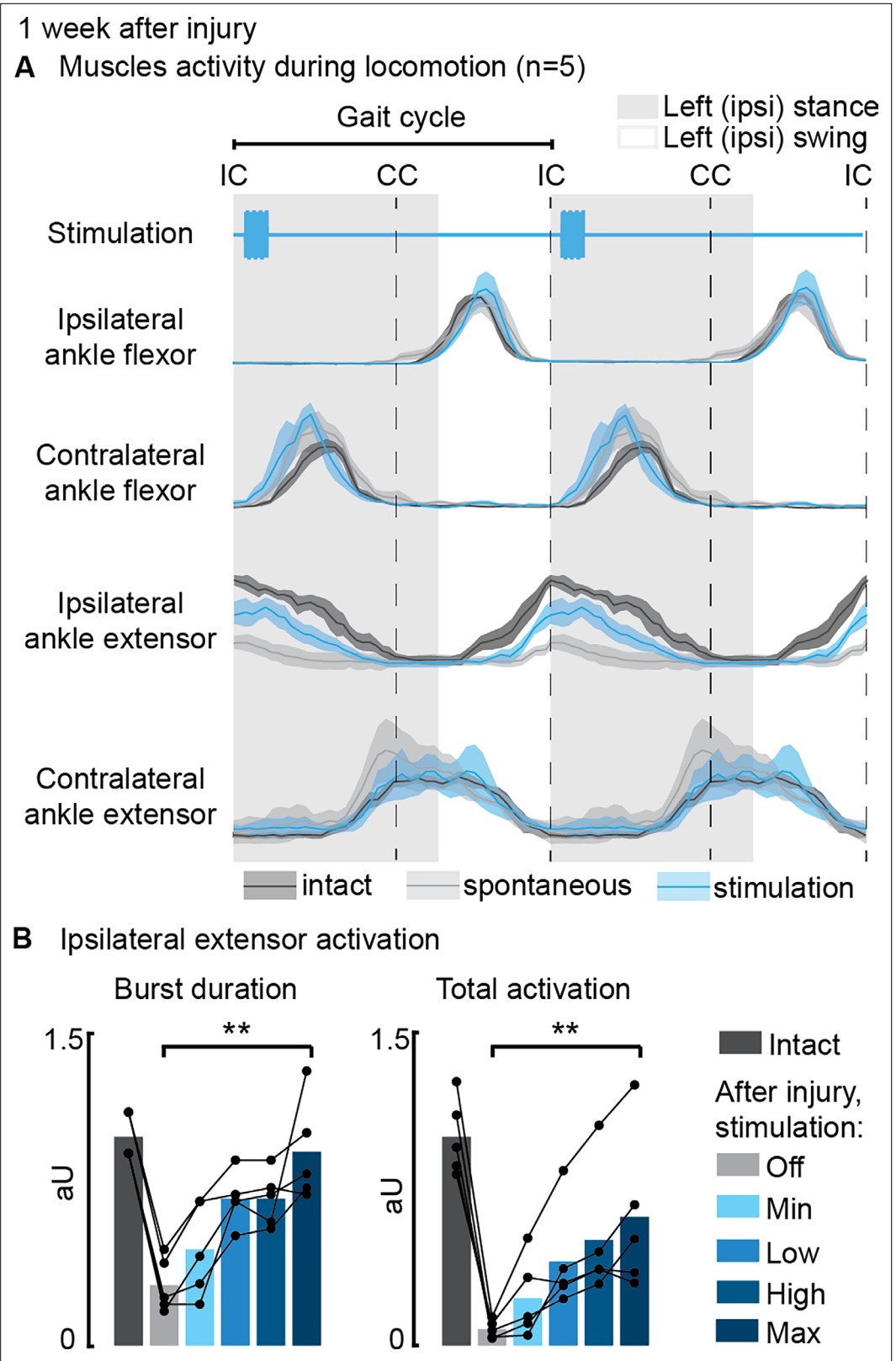

**Figure 4.** Phase-coherent intracortical stimulation reinstated ipsilateral extension muscle activity 1 week after injury (n=5 recordings for each muscle, from a total of 7 rats, with some muscles unavailable due to implant failure). (**A**) Electromyographic (EMG) envelopes during spontaneous locomotion before and after injury as well as phase-coherent stimulation after injury. Activities of the ipsilateral and contralateral ankle flexor (tibialis anterior) and

*Figure 4 continued on next page*

*Figure 4 continued*

ipsilateral and contralateral ankle extensor (medial gastrocnemius). The gait event references are reported as LC: left contact, RC: right contact. (**B**) Left medial gastrocnemius activity was modulated by the stimulation. The burst duration and the level of muscle activation were linearly increased with greater stimulation amplitudes. The data are represented as the mean ± SEM. **p<0.01. The hemisection profiles of the seven rats are identified by a blue star in *Figure 2A*.

The online version of this article includes the following source data for figure 4:

**Source data 1.** Source data for *Figure 4B*.

of the overall locomotor score recovery occurred (*Figure 6D*). Furthermore, 83 ± 7% of the increment in size of the ipsilateral map took place within the same time interval (*Figure 6B*).

For each individual rat, the trend of locomotor score improvement measured during the first 3 weeks after SCI correlated with both ipsilateral (VAF: 84 ± 3%) and contralateral (VAF: 82 ± 4%) hindlimb representations in the left cortex. This result is consistent with our previous finding on the contralateral representation in the right cortex, the one opposite to the lesion side (VAF: 70 ± 24%, *Bonizzato and Martinez, 2021*).

However, it is important to note that for any given day post-injury, the size of the left motor map across rats did not predict locomotor performance measured in an open field. The VAF was 3 ± 2% for ipsilateral transmission and 8 ± 4% for contralateral transmission (*Figure 6E–G*), compared to 56 ± 5% VAF when considering the opposite cortex (*Bonizzato and Martinez, 2021*). Additionally, the size of the left motor map did not correlate with lesion size (*Figure 6F*), unlike the opposite cortex, where a correlation was observed (*Bonizzato and Martinez, 2021*). Overall, with a left hemisection, both the left and right cortical motor maps expand in size with a similar temporal pattern to locomotor

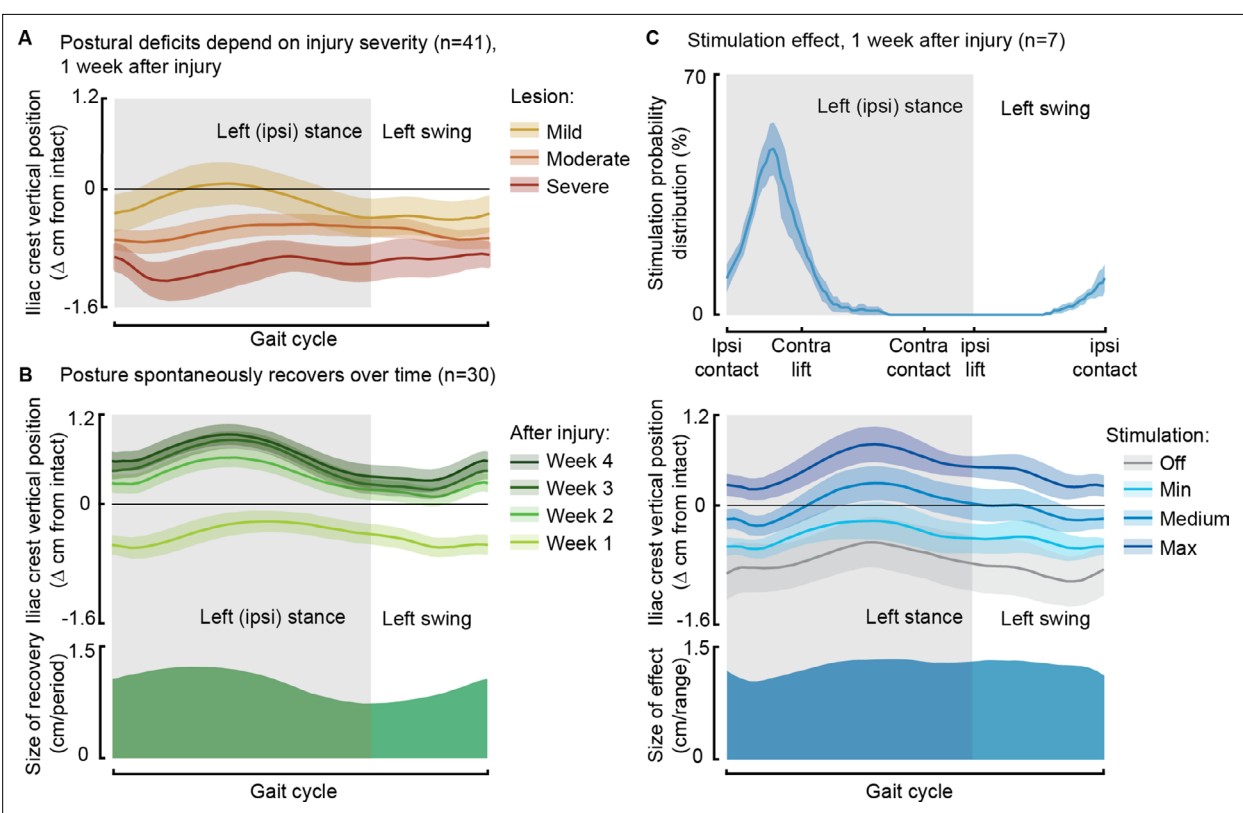

**Figure 5.** Phase-coherent stimulation improved posture 1 week after injury. Posture is shown as the height of the ipsilateral iliac crest during locomotion with respect to the spontaneous condition before injury. The data are represented as the mean ± SEM. (**A**) Postural deficits depend on injury severity. Rats with severe spinal cord injury (SCI) exhibit a weaker posture 1 week after injury. (**B**) Variation over 1 month of spontaneous recovery. Posture is raised and overcompensated. (**C**) Effect of phase-coherent stimulation 1 week after injury. Posture is increasingly raised with greater stimulation amplitudes. n=41, 30, or 7 rats, indicated in each panel.

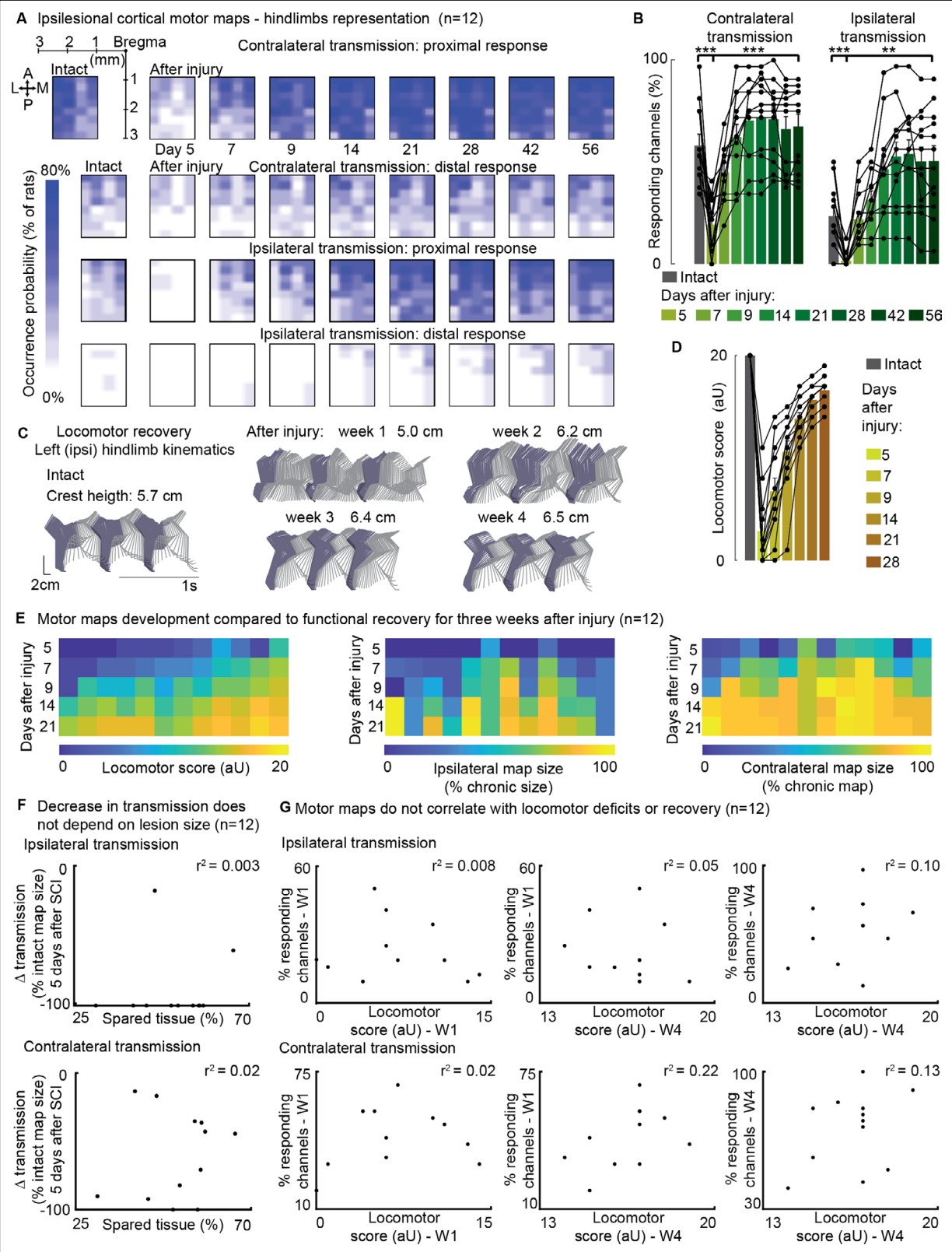

**Figure 6.** Ipsilateral motor representation of the affected hindlimb was increased in the ipsilesional motor cortex after injury but does not reflect functional recovery (n=12 rats). The term 'transmission' in figure indicates a quantification of the number of responding sites within the array, which is the surface whereby a stimulus transmission to the muscles resulted in a visible hindlimb contraction. (**A**) Awake cortical motor map representation before injury and up to 2 months after injury. The color intensity is proportional to the probability of evoking proximal/distal ipsi-/contra-lateral responses when

*Figure 6 continued on next page*

*Figure 6 continued*

stimulating a given site, across 12 animals. Bilateral representation of hindlimb movements increased over time compared to the intact condition. The top left sub-panel carries a representation of the electrode array positioning within the left motor cortex. (**B**) Quantification of responding channels from the intact condition and up to 2 months after injury. (**C**) Stick diagrams from treadmill locomotion and iliac crest height before injury and during the first 4 weeks after injury. (**D**) Quantification of locomotor score from the intact condition and up to 1 month after injury. (**E**) Corticospinal transmission and locomotor performance. Single rats (columns) are sorted by day 7 locomotor score. The same sorting is maintained for the three sub-panels. Absolute map size (i.e. responding sites on the implanted array, indicated as % of the final, chronic, size) did not correlate with higher or lower locomotor scores across rats, whereas the individual rats' trend of map size increase for 3 weeks paralleled locomotor recovery. (**F**) Lack of correlation between map size and lesion size 5 days after injury. An ipsilateral and contralateral decrease in transmission did not parallel the spared tissue at the lesion epicenter. (**G**) Lack of correlation between map size and locomotor score. Time points are reported as W1: week 1 and W4: week 4. Ipsilateral and contralateral transmission did not correlate with global locomotor recovery measured in an open field. Bars: mean of individual data points.

The online version of this article includes the following source data and figure supplement(s) for figure 6:

**Source data 1.** Source data for *Figure 6*.

**Figure supplement 1.** Assessment of correlation between motor maps and recovery of fine motor control evaluated as ladder score (n=12 rats).

**Figure supplement 1—source data 1.** Source data for *Figure 6—figure supplement 1*.

recovery. However, the absolute size of the left motor map does not predict lesion size or locomotor deficits, whereas the right motor map does.

When assessing the correlation between ipsilateral motor maps and skilled locomotor performance on the ladder task, we found that the return of distal representations within the ipsilateral motor maps correlated with the recovery of fine motor control 3 weeks after SCI (*Figure 6—figure supplement 1A*). This correlation disappeared 4 weeks after SCI and was not observed when including proximal movements. Additionally, the representation of the ipsilateral limb was notably variable between subjects.

## Ipsilateral neuromodulation of hindlimb flexion

Cortical control of hindlimb movements in behaving rats has been primarily associated with contralateral limb flexion and elevation (*Bonizzato and Martinez, 2021*; *Bonizzato et al., 2018*; *Brown and Martinez, 2021*; *DiGiovanna et al., 2016*; *Rigosa et al., 2015*). However, in our study, we observed a unique motor response in two out of the seven rats tested for phase-coherent stimulation (*Figure 7A*, *Supplementary file 1*). Specifically, in these two rats, tested 2 weeks after SCI, stimulation of specific array sites within a medial area of the hindlimb motor cortex (1.1 mm mediolateral from bregma, *Figure 7D*) preferentially evoked ipsilateral flexor responses (*Figure 7B–D*, rat#1: 3 channels with 271 ± 36% ipsilateral dominance, rest of responding channels 43 ± 4%, p=0.003, Wilcoxon rank-sum test, rat#2: 6 channels 452 ± 87%, all others 18 ± 4%, Wilcoxon rank-sum test, p=2E-4). The site with the highest ipsilateral dominance (rat#1: 327 ± 109%, rat#2: 692 ± 84%) was chosen to test the modulation of ipsilateral swing trajectories (*Figure 7B*). Stimuli delivered during the late ipsilateral stance resulted in kinematic modulation: step height (+133 ± 18, +99 ± 23%, p=[1E-4,0.001], Wilcoxon rank-sum test) and flexion velocity (+46 ± 19, +101 ± 19%, p=[0.01, 1E-4], Wilcoxon rank-sum test) increased linearly (rat#1: VAF=(90, 91)%, rat#2: VAF=(95, 86)%) with greater stimulation amplitudes (*Figure 7E*, *Figure 7—video 1*). As a result, dragging was immediately alleviated (–46±6, –100%, p=[1E-6, 7E-4], Wilcoxon rank-sum test). This result was unique for ipsi-dominant cortical sites; no other tested electrode produced ipsilateral flexion facilitation (see *Figure 3E*). The sites produced a similar functional effect as contralesional cortical stimulation (*Bonizzato and Martinez, 2021*).

## The ipsilesional motor cortex does not modulate ipsilesional movements through the homologous motor cortex

In three rats we combined thoracic unilateral SCI with a surgical ablation of the contralateral motor cortex (*Figure 8A*) to determine whether contralateral cortical networks are necessary to ipsilateral hindlimb modulation. In all three rats, modulation of leg extension was readily obtained through phase-coherent ipsilateral cortical neurostimulation 1 week after SCI. All three rats displayed postural deficits that were immediately alleviated by phase-coherent stimulation of the ipsilesional motor cortex (*Figure 8B and C*).

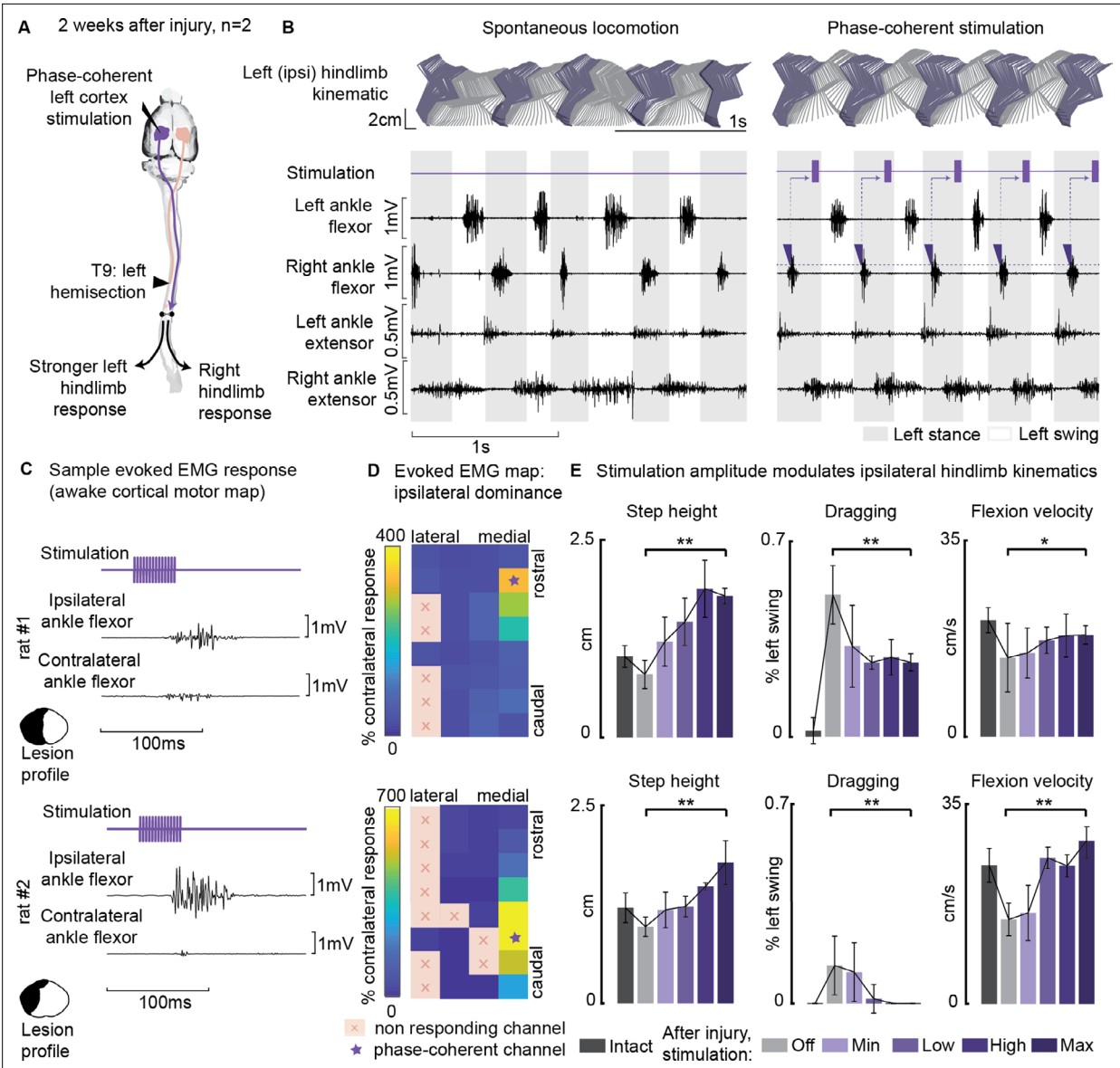

**Figure 7.** Ipsilesional motor cortex stimulation modulated ipsilateral hindlimb movements 2 weeks after injury (n=10 steps, experiment repeated in two rats). (**A**) A schematic representation of the injury and neurostimulation model. After lateral hemisection, ipsilesional motor cortex stimulation evoked ipsilateral responses. (**B**) Stick diagrams and electromyographic (EMG) activity during spontaneous locomotion and phase-coherent stimulation. The stimulation was triggered by contralateral ankle flexor activation and was delivered during the late ipsilateral stance. (**C**) Samples of single train stimulation of specific channels that preferentially evoked ipsilateral muscle activation in two different animals. (**D**) Ipsilateral dominance of EMG responses. Awake cortical motor maps were obtained as a ratio between ipsilateral and contralateral tibialis anterior activation. Channels that presented an ipsilateral preference were located in the most medial region of the map and identified by a star. (**E**) Phase-coherent stimulation modulated ipsilateral kinematics and reduced the foot drop deficit. Ipsilateral step height, flexion speed, and dragging alleviation linearly increased with greater stimulation amplitudes. Two subjects are presented independently, 10 steps per condition. The hemisection profiles of the two rats are identified by a purple star in *Figure 2A*. The data are represented as the mean ± SEM. *, ** p<0.05 and <0.01, respectively.

The online version of this article includes the following video and source data for figure 7:

**Source data 1.** Source data for *Figure 7D and E*.

**Figure 7—video 1.** Cortical neuroprosthesis-mediated control of ipsilateral hindlimb flexion.
https://elifesciences.org/articles/92940/figures#fig7video1

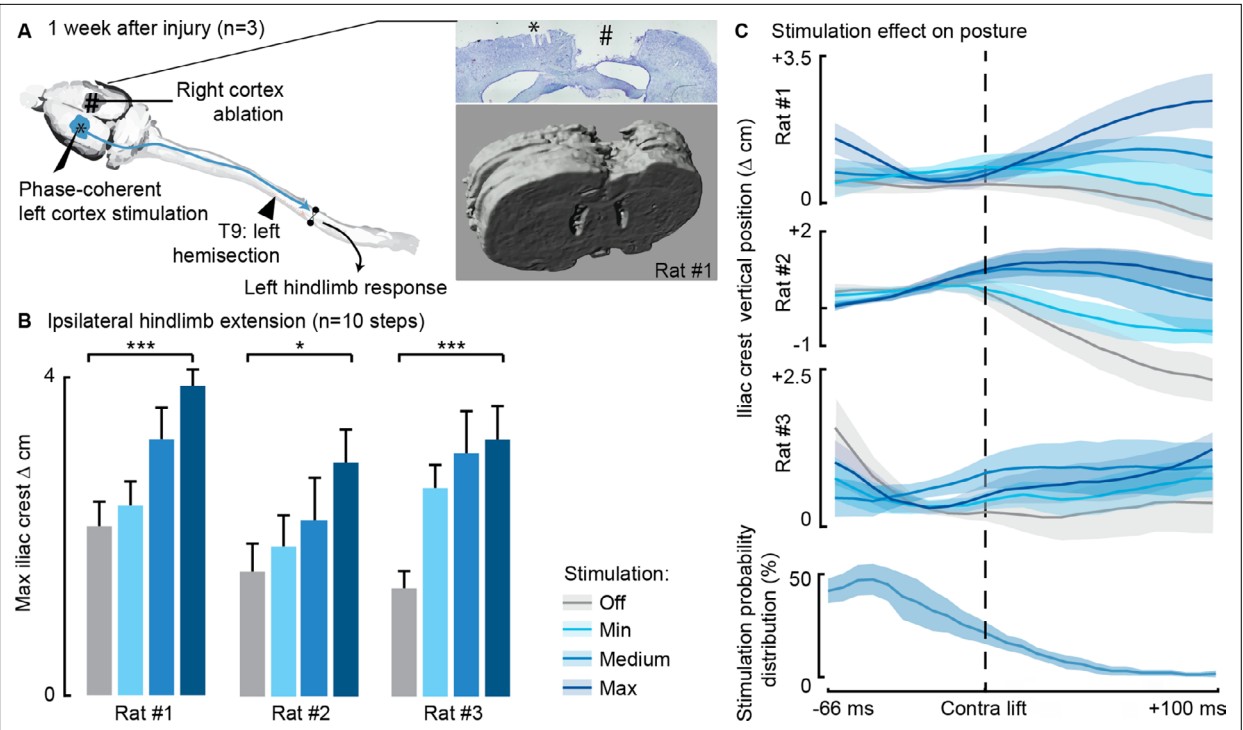

**Figure 8.** Phase-coherent stimulation of the ipsilesional motor cortex alleviated locomotor deficits even after ablation of the contralateral motor cortex (n=3 rats). (**A**) Schematic representation of the injury and neurostimulation model. After lateral hemisection, ipsilesional motor cortex stimulation evoked ipsilateral responses, even when the contralateral motor cortex was ablated. Right inset: top, Cresyl violet staining of a coronal brain slice of rat #1. *, electrode traces in the left cortex. #, right cortex ablation. (**B–C**) Phase-coherent stimulation modulated ipsilateral kinematics. Posture linearly increased with greater stimulation amplitudes. Three rats are presented independently, 10 steps per condition. The data are represented as the mean ± SEM. *, *** p<0.05 and <0.001, respectively.

The online version of this article includes the following source data for figure 8:

**Source data 1.** Source data for *Figure 8B*.

## Cortical neuromodulation of hindlimb alternated rhythms

We next investigated whether long-train intracortical stimulation in awake, resting rats could evoke complex multi-modal motor responses (*Graziano et al., 2002*) and whether the effects on hindlimb movement are bilateral. The stimulation lasted 250 ms, approximately matching the time scale of locomotor movement preparation and initiation (*Bonizzato and Martinez, 2021*). In six intact rats, we found that long-train stimulation of one motor cortex evoked locomotor-like rhythms (*Figure 9A and B*, *Figure 9—video 1*), characterized by bilateral alternated whole-leg movements.

Subsequently, we assessed whether 1 week after unilateral hemisection SCI, long-train stimulation of the ipsilesional motor cortex could induce bilateral rhythms. We observed that in half of the tested rats with more severe injuries and lower ladder crossing performance, bilateral alternated locomotor-like rhythms did not emerge immediately after injury. However, by week 2 or 3 post-injury, these bilateral rhythms returned (*Figure 9C*). In contrast, the remaining three rats with less severe injuries exhibited bilateral alternated hindlimb rhythms when receiving ipsilesional cortical stimulation as early as 1 week after injury (*Figure 9D*). Classically, studies of cortical control and recovery of movement are often conducted under ketamine sedation (*Brown and Martinez, 2018*; *Nudo et al., 1996b*). To illustrate the well-known absence of rhythmic hindlimb activity after ketamine sedation, we tested and recorded four intact rats before and after ketamine injection, confirming the suppression of rhythmic hindlimb responses (*Figure 9E–F*).

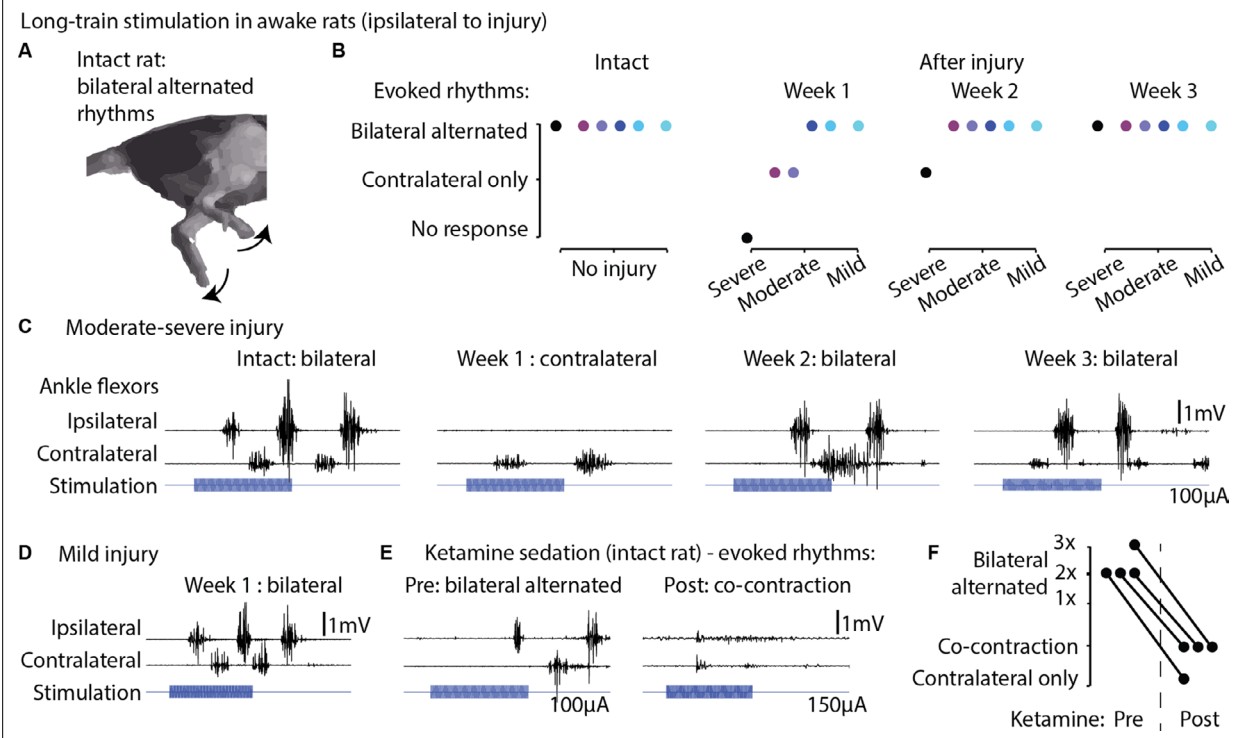

**Figure 9.** Long-train intracortical stimulation in awake rats elicited alternated bilateral rhythms. (**A**) Schematic representation of the locomotor-like rhythmic movements evoked by long-train (250 ms) cortical stimulation (amplitude 100 µA). Evoked rhythms are characterized by alternated hindlimb movements. (**B**) In six intact rats, stimulation of the left motor cortex generated bilateral alternated hindlimb rhythms. After spinal cord injury (SCI), rats are sorted by injury severity, using their ladder score at week 1 for ranking. One week after injury, long-train cortical stimulation failed to evoke bilateral alternated rhythms in half of the cohort. In two of these rats, contralateral rhythms were still present and bilateral alternated rhythms were recovered by week 2. In the most severe rat, contralateral-only rhythms were evoked on week 2 and bilateral alternated rhythms on week 3. For the remaining half of the cohort, long-train cortical stimulation recruited bilateral alternated rhythms at all tested time points. (**C**) Stimulus-synchronized ankle flexor electromyographic (EMG) traces from one rat with a moderate-severe injury, showing loss (week 1) and following recovery (weeks 2–3) of ipsilateral evoked hindlimb rhythms. (**D**) Stimulus-synchronized EMG trace from one rat with mild injury, showing that bilateral alternated evoked rhythms are preserved at week 1. (**E**) Stimulus-synchronized EMG trace from four intact rats before and after ketamine sedation, showing transient loss of bilateral alternated rhythms. (**F**) Loss of bilateral alternated rhythms in four rats after ketamine sedation. 1X, 2X, 3X: number of complete repetitions of alternating movements produced by long-train cortical stimulation (amplitude 150 µA).

The online version of this article includes the following video and source data for figure 9:

**Source data 1.** Source data for *Figure 9B and F*.

**Figure 9—video 1.** Long-train cortical stimulation recruits spinal locomotor circuits.

https://elifesciences.org/articles/92940/figures#fig9video1

# Discussion
## A cortical neuroprosthesis facilitates the control of ipsilateral hindlimb extension

In this study we demonstrated that, after a lateralized SCI, the ipsilesional motor cortex (with most of its crossed efferences preserved) played a prominent role in controlling bilateral hindlimb movements. Our ipsilateral cortical neuroprosthesis effectively alleviated SCI-induced locomotor and postural deficits across different levels of injury severity (*Figure 2*, *Supplementary file 1*), even after removal of the homologous motor cortex (*Figure 8*). The lateralized lesion model and phase-coherent cortical stimulation revealed functional ipsilateral motor control. The evoked movement was characterized by contralateral hindlimb flexion accompanied by simultaneous ipsilateral hindlimb extension. Thus, the ipsilesional motor cortex can activate and influence bilateral lumbar synergistic networks through descending connections spared by the injury. We propose that the acute expression of this bilateral synergy (1 week after injury) is compatible with an adaptive or compensatory upregulation of

pre-existing functional networks after SCI. Rapid onset of postural compensation is also displayed behaviorally by rats during the same timeframe (*Figure 5B*). Although this outcome may reflect the participation of several supralesional networks, lateralized injuries highlight the role of the ipsilesional motor cortex in voluntary postural and weight-bearing adjustments. We hypothesize that this phenomenon is indicative of the necessity to preserve the functional role of the motor cortex in modulating contralateral step height during locomotion. In the absence of appropriate support from the opposite hindlimb due to the injury, the ability to elevate the foot would be compromised. Therefore, cortex-driven descending pathways may increase the excitatory transmission to ipsilesional extensor networks, thus facilitating the restoration of appropriated hindlimb support and precise functional control of contralateral step height. We postulate that this may represent either a demonstration of redundancy emerging with the lesion, and/or a specific compensatory strategy.

## Ipsilesional motor map progression after SCI did not correlate with spontaneous recovery

After a unilateral cortical injury, plastic changes are observed in the opposite hemisphere (*Axelson et al., 2013*; *Dancause et al., 2005*; *Mansoori et al., 2014*; *Rehme et al., 2012*; *Shimizu et al., 2002*; *Strens et al., 2003*; *Witte et al., 2000*). Laterally unbalanced SCIs induce dynamic changes in the contralesional and ipsilesional motor cortex, which may participate in functional recovery or compensation mechanisms (*Bonizzato and Martinez, 2021*; *Brown and Martinez, 2018*; *Brown and Martinez, 2021*; *Ghosh et al., 2010*; *Ghosh et al., 2009*; *Nishimura et al., 2007*; *Schmidlin et al., 2005*). The relationship between map plasticity and motor recover is, however, complex: motor maps are static representations of a dynamic and modifiable system that is under the influence of experience (*Milliken et al., 2013*; *Nudo et al., 1996a*; *Oza and Giszter, 2015*; *Singleton et al., 2021*) and interconnected circuits state (*Ethier et al., 2007*). Some studies have shown that the reorganization of motor maps does not correlate with the time-course of behavioral recovery (*Eisner-Janowicz et al., 2008*; *Nishibe et al., 2015*; *Plautz et al., 2023*; *Wang et al., 2010*). In this study, we derived cortical maps in awake animals to investigate the time-course of ipsilateral transmission between the motor cortex and spinal circuits. The main advantages of awake mapping are twofold: first, this technique allows to longitudinally track motor cortex plasticity in the same animal; second, awake mapping unveils non-pyramidal transmission, which is suppressed by ketamine anesthesia (*Bonizzato and Martinez, 2021*). We tracked the progression of motor representation of both hindlimbs in the ipsilesional motor cortex and found that in all rats, corticospinal transmission significantly decreased after injury (*Figure 6B*), independently from the subject-specific size of the injury (*Figure 6F*). This finding is consistent with a major loss of connectivity, including damage to the uncrossed ventral CST (*Weidner et al., 2001*) and ipsilateral cortico-reticulo-spinal transmission (*Bonizzato and Martinez, 2021*). The loss of excitability quickly recovered within 2 weeks (*Figure 6B*), with a return of corticospinal transmission consistent with the upregulation of the descending pathways spared by the injury. However, the subject-specific size of cortical motor maps did not correlate with the behaviorally expressed global motor performance measured in an open field (*Figure 6G*). Conversely, we previously showed that contralesional cortical map size across subject correlated with locomotor recovery measured in an open field (*Bonizzato and Martinez, 2021*). Comparison of these two results suggested that recovery of hindlimb movement after SCI may be more tightly connected to changes in the contralateral cortical motor representation rather than the ipsilateral cortical motor representation, even in fully lateralized thoracic injuries, which disproportionally affect the crossed projections from the contralateral motor cortex. Nevertheless, we found that the return of ipsilateral distal transmission paralleled the recovery of fine motor control assessed in the ladder task, but this effect was transitory and restricted to the third week after SCI (*Figure 6—figure supplement 1*). This is in line with our previous work showing that acute cortical inactivation immediately reinstated bilateral hindlimb deficits on the ladder task but only 3 weeks after SCI (*Brown and Martinez, 2018*). These combined results suggested that, although not a precise predictor of motor performance, the return of bilateral corticospinal transmission from the ipsilesional motor cortex after SCI is an important excitatory drive that supports bilateral skilled hindlimb movement. Given that bi-cortical interactions in shaping descending commands are established (*Brus-Ramer et al., 2009*), and considering the changes we observed in ipsilesional motor cortex excitability, the potential role of the ipsilateral cortex in mediating or supporting functional descending commands from the contralateral cortex - particularly in the immediate increase in flexion

of the affected hindlimb and the long-term recovery of functional control (*Bonizzato and Martinez, 2021*) - warrants further investigation.

## A cortical neuroprosthesis facilitates the control of ipsilateral hindlimb flexion

We observed a unique case of ipsilateral hindlimb flexion modulation in two rats that deserves specific consideration. These rats had arrays positioned at precisely the same brain coordinates ([1.1 mm posterior, 1.1 .mm lateral] from bregma) and depth (1.5 mm). However, their lesion profiles were substantially different (see *Figure 2A*, purple stars and *Supplementary file 1*). In one rat, all descending tracts were interrupted on one side while in the other rat, the lesion spared CST pathways and non-pyramidal ventral tracts. Interestingly, the remaining five rats (see *Figure 2A*, blue stars) used to test immediate modulation of movement under cortical stimulation did not exhibit ipsilateral hindlimb flexion, despite having variable lesion profiles. Thus, we did not find a relationship between spared pathways on the lesioned side and the capacity to neuroprosthetically achieve ipsilateral flexion modulation. Additionally, all rats had their right hemicord preserved, suggesting that pathways traveling on the intact side are also unlikely to be involved in the observed results in two out of seven rats.

The localization of the specific channels closest to the interhemispheric fissure (*Figure 7D*) may suggest the involvement of transcallosal interactions in mediating the transmission of the cortical command generated in the ipsilateral motor cortex (*Brus-Ramer et al., 2009*). While ablation experiments (*Figure 8*) refute this hypothesis for ipsilateral extension control, they do not conclusively determine whether a different efferent pathway is involved in ipsilateral flexion control in this specific case.

As an alternative hypothesis to explain the low incidence of rats presenting electrodes evoking ipsilateral flexion, one might consider inter-individual variability in cortical motor maps. We know from our previous work that the localization of hindlimb motor maps varies between rats (around 0.5 mm medial from bregma between rats) (*Bonizzato and Martinez, 2021*; *Brown and Martinez, 2018*; *Brown and Martinez, 2021*). Since the channels generating ipsilateral flexion were found to be the most medial in the map, it is possible that more of these responses could be obtained if the array was positioned more medially. However, performing the craniotomy and inserting the array at such coordinates would be challenging due to the risk to damage the superior sagittal sinus and inducing hemorrhage.

Further experiments are necessary to understand the mechanism(s) underlying this unconventional instance of cortical control of movement and whether they are mediated by cortical efferences, transcallosal communication, brainstem relays, or spinal networks. A compelling research question arising from these results is whether similar findings can be found in the primate motor cortex.

## A cortical neuroprosthesis unveiled ipsilateral functional control of movement

Numerous hypotheses have been proposed to explain ipsilateral motor cortical activity during movement, and our study contributes to this ongoing debate by establishing specific causal links in brain-behavior interactions (*Silvanto and Pascual-Leone, 2012*). These hypotheses include:

1. An abstract, limb-independent representation of movement (*Porro et al., 2000*).
2. The presence of an efference copy of signals generated by the contralateral motor cortex (*Ganguly et al., 2009*).
3. The existence of uncrossed descending connectivity (*Brinkman and Kuypers, 1973*; *Nathan et al., 1990*; *Rosenzweig et al., 2009*; *Stecina and Jankowska, 2007*; *Weidner et al., 2001*).
4. Bilateral termination of crossed descending connectivity (*Becker et al., 2010*; *Lacroix et al., 2004*; *Rosenzweig et al., 2009*).
5. Distributed (*Ames and Churchland, 2019*; *Cisek et al., 2003*; *Li et al., 2016*) or overlapping (*Gazzaniga, 2000*; *Merrick et al., 2022*; *Parsons et al., 1998*; *Schaefer et al., 2007*; *Volpe et al., 1982*) motor cortical computations across the two hemispheres.

Our study provides evidence of cortical-mediated control of functional, complex, and diverse ipsilateral movements in rats, even after the ablation of the homologous motor cortex. These findings challenge the view that ipsilateral motor cortex activity is solely epiphenomenal, a purely abstract representation, or a mere efference copy. Importantly, our results indicate that the complex bilaterality of cortical descending projections, as suggested in hypothesis (4), persists and does not rely solely on

uncrossed pathways, as lateralized injury completely abolished all uncrossed descending connectivity in some animals while the observed effects persisted. This complexity highlights the intricate nature of neural control of movement and raises questions about the interplay of various neural pathways in motor control.

This hypothesis gains further supported from the observation that in rats, ipsilesional cortical inactivation immediately reinstates leg control deficits 3 weeks after hemisection (*Brown and Martinez, 2018*). While the mechanisms through which the ipsilateral motor cortex influences spinal circuits are likely multifaceted, several studies have proposed that the upregulation of indirect cortico-reticulospinal pathways, which are partially spared in our rats, may serve as a neural substrate for transmitting cortical drive (*Asboth et al., 2018*; *Bonizzato and Martinez, 2021*). Following hemisection SCI in rats, it has been proposed that the ipsilesional reticular formation could influence locomotor functions either through reciprocal connections with its contralesional counterpart (*Zörner et al., 2014*) or detour pathways involving relay interneurons within the spinal cord (*Cowley et al., 2008*). Given that the brainstem's reticular formation is known to control bilateral flexor and extensor synergies during locomotion, intracortical stimulation may gate the modulation of flexion and extension-related outputs of pattern generation in a phase-dependent manner through preserved corticospinal or cortico-reticulospinal pathways (*Bretzner and Drew, 2005*; *Drew, 1991*; *Drew and Rossignol, 1984*; *Dyson et al., 2014*; *Fortier-Lebel et al., 2021*; *Lemieux and Bretzner, 2019*). The recruitment of ipsilateral synergies through intracortical stimulation after spinal hemisection likely also involves spinal plasticity and changes in excitability (*Martinez et al., 2011*; *Martinez et al., 2012*; *Martinez et al., 2013*). Spinal circuits contain sets of couple oscillators, one for flexion and one for extension, which are reciprocally connected but independently regulated (*McCrea and Rybak, 2008*). After a spinal hemisection, activity within extensors decreases on the lesion side while mirror effects occur in the intact limb, altering the balance between flexion and extension rhythms generators (*Brown and Martinez, 2021*; *Martinez and Rossignol, 2013*). These functional changes after hemisection are highly sensitive to activity within remaining pathways on the intact side of the spinal cord (*Martinez et al., 2012*), including residual corticospinal projections (*Brown and Martinez, 2021*; *Brustein and Rossignol, 1998*; *Górska et al., 1993*). Therefore, increasing ipsilateral cortical drive with intracortical stimulation during locomotion may have recruit new synergies or uncover novel modes of locomotor control. Given that corticospinal-interneuronal connections primarily mediate movements and reflexes in adult rodents and primates (*Lemon, 2008*), corticospinal inputs mainly influence bilateral motor synergies through these interneurons. These interneurons are significantly affected by sensory inputs that are crucial in reflex pathways, essential for coordinating limb movements and maintaining posture. Mutual inhibitory interactions between interneuronal networks that control opposite actions, such as Ia inhibition of either flexors or extensors (*Hultborn et al., 1976a*; *Hultborn et al., 1976b*), as well as commissural interneurons connecting the left and right sides (*Harrison et al., 1986*), modulate motoneuronal activity and work together to coordinate movements between the left and right limbs and to manage crossed reflex loops. These circuits are sensitive to supraspinal inputs, and depending on the activation of these pathways, they may support various motor outputs, including motor learning, unskilled, and skilled movements.

However, it is essential to note a distinction between the specificity of the ipsilateral cortical role in functional motor control and motor recovery. Control and recovery are distinct physiological processes. While our results demonstrate the ability of the motor cortex to specifically control ipsilateral extension (*Figures 1 and 3–5*) and flexion (*Figure 7*) movements linearly with increasing stimulation amplitudes, we did not observe a clear predictive relationship between changes in motor transmission from the ipsilesional cortex and functional measures of global locomotor recovery assessed in an open field (*Figure 6*). In contrast, this type of analysis yielded positive results when applied to the contralateral motor cortex (*Bonizzato and Martinez, 2021*). Further investigation could clarify how corticospinal neurons contribute to the recruitment of ipsilateral spinal networks and the broader implications for motor behavior, particularly in the context of neuroprosthetic interventions and SCI recovery.

## Long-train cortical stimulation recruits spinal locomotor circuits

The brief duration of the stimulus train typically used in phase-coherent stimulation experiments may limit the display of complete and coordinated movements that can be evoked and modulated by cortical networks when activated for longer periods (*Baldwin et al., 2017*; *Baldwin et al., 2018*;

*Bonazzi et al., 2013*; *Brown et al., 2023*; *Brown and Teskey, 2014*; *Graziano et al., 2002*; *Halley et al., 2020*; *Mayer et al., 2019*). This limitation contrasts with the endogenous movement initiation process, which operates over hundreds of milliseconds (*Bonizzato and Martinez, 2021*). To address this, we employed long-train cortical stimulation in resting animals to elicit complex locomotor-like rhythms. These evoked movements exhibited high coordination bilaterally across the entire hindlimb system. While afferent inputs are known to play a crucial role in spinal locomotion (*Alluin et al., 2015*; *Barthélemy et al., 2007*; *Sławińska et al., 2012*), our study demonstrates that unilateral cortical drive can activate spinal locomotor circuits, leading to the generation of alternated 'air-stepping' in awake rats even in the absence of cutaneous interaction with a ground surface. Furthermore, we observed that thoracic hemisection initially restricts the effects of cortical excitation to the unilateral genera-tion of spinal rhythms. This suggests that cortical projections recruit independent rhythm-generating spinal units, which can be side-specific (*Grillner and Wallén, 1985*). However, the recovery of bilateral alternated rhythms within 2–3 weeks after hemisection implies changes within the spinal circuitry below the lesion, possibly mediated by the persistent interaction between commissural interneu-rons and efferences responsible for corticospinal transmission (*Gossard et al., 2015*; *Martinez et al., 2011*). The role of supraspinal drive on spinal locomotor circuits has been previously discussed with respect to 'fictive' locomotion (decerebrate) preparations. In the cat, pyramidal stimulation was found to reset the locomotor rhythm by initiating bursts of activity in either extensor (*Leblond et al., 2001*) or flexor muscles (*Orlovsky, 1972*), but repetitive burst stimulation was required to evoke repeated hindlimb responses and structured them into a rhythm. This falls short to the rhythms-evoking capacity we demonstrated through long-train cortical stimulation in awake rats.

## Ipsilateral cortical control of movement

Our findings reveal that brief, phasic cortical stimulation generates specific cortical commands, either for flexion or extension, and these commands are transmitted to both hindlimbs when applied during the appropriate phase of the locomotor cycle. In contrast, prolonged cortical stimulation activates spinal locomotor circuits, effectively converting unilateral cortical neuromodulation into a bilateral alternating output. This transformation demonstrates the complex executive relationship between the rodent motor cortex and spinal networks responsible for cortical initiation and modulation of ongoing movement. This interaction allows for a bilateral efferent transmission, effectively integrating and regulating spinal states. Movement generation involves the coordinated activity of distributed cortical, subcortical, brainstem, and spinal networks, each strongly interconnected with its contralat-eral counterpart. Multiple cortical networks contributing to movement generation have been shown to activate in a limb-independent manner. In the dorsal stream of visuomotor processing, for instance, the posterior parietal cortex contributes to grasping (*Kermadi et al., 2000*) or locomotor movements such as obstacle avoidance (*Andujar et al., 2010*), with neurons responding to both left and right limb movements predominating. Similarly, premotor cortical areas contain neurons that become activated during ipsilateral movement (*Cisek et al., 2003*; *Kermadi et al., 2000*; *Michaels and Scherberger, 2018*). Our results suggest that this bilaterality is not extinguished in the cortical line of sensorimotor integration. Instead, it is selectively preserved in the functional network properties of the primary motor cortex, the ultimate cortical actuator of movement.

## Cortical neuroprostheses

These findings, in addition to shedding light on the intricate ipsilateral control of movement in rats, carry promising translational implications for the future development of neuroprosthetic solutions. Our previous work has demonstrated that phase-dependent cortical stimulation applied to the contrale-sional motor cortex immediately ameliorates dragging deficits following SCI by specifically enhancing contralateral hindlimb flexion (*Bonizzato and Martinez, 2021*; *Duguay et al., 2023*). Given that ipsilesional cortical stimulation induces a bilateral synergy, leading to the improvement of the affected limb's extension, this approach has the potential to effectively complement contralesional cortical stimulation. The ultimate aim would be to promptly reverse both postural and locomotor deficits associated with lateralized lesions. As per our previous study (*Bonizzato and Martinez, 2021*), future work could embed ipsilateral stimulation into rehabilitative training and evaluate its long-term impact over locomotor recovery. Since ipsilesional cortical stimulation immediately alleviated motor deficits in rats, and effects were maintained after the ablation of the contralateral cortex, it may also promote

more efficient movement execution in individuals with lateralized SCI or hemiparesis due to cortical or subcortical stroke. Improved motor performance may lead to a broad range of potential benefits, including better and more sustained access to activity-based training. A limitation of this potential strategy is the invasive nature of the intracortical interface utilized in the rats. Less invasive solutions exist including transcranial magnetic stimulation, which requires further targeted research since (1) it has not yet been tested as a 'priming' agent for movement in the subacute phases of neurotrauma (*Smith and Stinear, 2016*) and (2) it is usually intended as an inhibitory agent for the non-lesioned cortex (*Nowak et al., 2009*), in line with the interhemispheric inhibition stroke model. A clear trade-off between invasiveness and efficacy of neurostimulation techniques needs to be established to determine which set of neurostimulation methods holds the potential to improve the generation of cortical motor commands in individuals with neurotrauma.

## Materials and methods

### Experimental model and subject details

#### Animals

All procedures adhered to the guidelines established by the Canadian Council of Animal Care and received approval from the Comité de déontologie de l'expérimentation sur les animaux (CDEA, protocol #17-083), the animal ethics committee at Université de Montréal. A total of 16 female Long-Evans rats (Charles River Laboratories, line 006, weighing between 270 and 350 g, as detailed in *Supplementary file 1*) were utilized for this study. Additional rats (n=25) were included in the analysis of spontaneous postural changes following injury (*Figure 5A and B*).

Following an acclimatization period and habituation to handling, the rats were trained to ambulate on a motorized treadmill using positive reinforcement in the form of food rewards. Prior to surgery, the rats were housed by groups of three, but after implantation, they were housed individually. The blinding approach was not applicable in this case, as kinematic analysis was automatically conducted by DeepLabCut. The output data underwent curation to rectify any detection errors, and such corrections accounted for less than 0.5% of the conditioned points.

#### Study design

The number of animals used in this study was determined through a power analysis. The primary objective of this study was to assess the immediate effects of phase-coherent intracortical stimulation on modulating ipsilateral movements both before and after a unilateral SCI. The specific aims were to use ipsilesional motor cortex stimulation to enhance the extension/stance phase and to improve weight support of the affected hindlimb after unilateral SCI. At the outset of the study, a pilot experiment involving two animals revealed that ipsilesional phase-coherent intracortical stimulation resulted in an increase of over 80% in the duration of the contralateral stance phase when compared to intra-subject variability. Based on this finding, a power analysis was conducted, which estimated a 97% probability of achieving statistically significant results (α=0.05) with a sample size of n=5 subjects and a 99% probability with n=6 subjects (one-sided, paired t-test). Initially, we characterized a total of six intact animals. For subjects with SCI, we expanded the sample size to n=7 to ensure an adequate power for electromyographic (EMG) investigations. Subsequently, after excluding data from recordings that exhibited poor signal quality, we conducted an EMG analysis with five animals for each muscle.

### Method details

#### Surgical procedures

All surgical procedures were performed under isoflurane general anesthesia. Lidocaine (2%) was administered at the incision sites for local anesthesia. Analgesic (buprenorphine) and antibiotic (Baytril) medications were administrated for 3–4 days following surgery to ensure the animals' comfort and prevent infection.

During the initial surgery, we implanted the EMG electrodes and the intracortical array. Differential EMG wires were inserted into the left and right tibialis anterior and medial gastrocnemius muscles, while common ground wires were subcutaneously placed around the torso. A craniotomy was performed, and the dura mater was removed from the left motor cortex hindlimb area. Subsequently, a Tucker-Davis Technologies 32-channel array (consisting of 8 rows and 4 columns, measuring

1.125×1.75 mm$^2$) was inserted into cortical layer V at a depth of 1.5 mm, with the top-right site of the array positioned at coordinates (1.1 mm posterior, 1.1 mm lateral) from bregma. The EMG connector and intracortical array were then embedded in dental acrylic and secured on the head using four screws.

In the second surgery, SCI was induced in the rats. This involved performing a partial T9 laminectomy and using 2% lidocaine to reduce spinal reflexes. Subsequently, a left spinal cord hemisection procedure was performed as described by *Brown and Martinez, 2019*. In cases where rats experienced difficulty with micturition, their bladders were manually expressed for several days following the injury until they regained spontaneous control of micturition.

## Behavioral assessments

The motor performance of the rats was assessed using three tasks: (1) ladder crossing, (2) open-field, and (3) treadmill.

1.  Recorded at a frame rate of 100 frames per second while crossing a horizontal ladder measuring 130 cm in length, with rungs (3 mm diameter) regularly spaced at 2 cm intervals. In each session, trials involving consecutive steps were analyzed, and the results of five trials per rat were averaged. Each trial consisted of approximately 10 steps. The scoring system was based on the foot fault score (*Metz and Whishaw, 2002*). Seven days after the induction of the lesion, the performance was used as a reference to classify the severity of the animal's injury. Injuries were categorized based on the number of partial or correct paw placements on the rungs relative to the total number of steps (referred to as paw placements). Consequently, injuries were classified as follows: (i) mild (left hindlimb>20% paw placement), (ii) moderate (left hindlimb<20% paw placement and right hindlimb>75% paw placement), and (iii) severe (bilateral deficit, right hindlimb<75% paw placement).

2.  To assess the spontaneous recovery of global locomotor and postural abilities, rats underwent evaluation in an open field using an adapted version of a neurological scoring scale originally developed for assessing locomotor function after cervical SCI (*Brown and Martinez, 2019*; *Martinez et al., 2009*). During this test, rats were recorded at a frame rate of 30 frames per second while engaging in 4 min of spontaneous locomotion within a circular Plexiglas arena with a diameter of 96 cm and wall height of 40 cm, featuring an anti-skid floor. The locomotor score was assigned based on the Martinez scale, which took into account specific parameters: (i) articular movement amplitude of hip, knee, and ankle (0=absent, 1=slight, 2=normal); (ii) stationary and active weight support of the limb (0=absent, 1=present); (iii) digit position of hindlimb (0=flexed, 1=atonic, 2=extended); (iv) paw placement at initial contact (0=dorsal, 1=internal/external rotation, 2=parallel); (v) paw orientation during lift (1=internal/external rotation, 2=parallel); (vi) movement during swing (1=irregular, 2=regular); (vii) coordination between the fore- and hindlimb (0=absent, 1=occasional, 2=frequent, 3=consistent); and (viii) tail position (0=down, 1=up) for a maximum of 20 points.

3.  The treadmill task was employed to assess the effects of stimulation on hindlimb kinematics, posture, and muscular activity. Each trial involved the analysis of 10 consecutive steps, with the treadmill set at a speed of 23 cm/s. Kinematics were recorded at a rate of 119.2 Hz using six reflective markers placed on key anatomical points, including the iliac crest, trochanter, knee, fifth metatarsal, and fourth toe tip. The kinematic data were processed using DeepLabCut (*Mathis et al., 2018*) and underwent manual curation to correct any misidentifications. Gait analysis was subsequently performed to identify important locomotor performance parameters. *Stance* was defined as the phase of the gait between foot contact and the subsequent lift, while *swing* was defined as the phase between lift and the following foot contact. *Swing asymmetry* (left vs right) was defined as 1-(SwLeft/SwRight). *Stance asymmetry* (left vs right) was defined as 1-(StLeft/StRight). SwLeft or SwRight, StLeft or StRight indicate the duration of swing and stance phases, respectively. Negative values indicated that the left leg had a shorter duration than the right, while positive values indicated the opposite. *Flexion velocity* referred to the maximum vertical speed of the foot during hindlimb flexion occurring in the swing phase. *Step height* was calculated by subtracting the average vertical position of the foot during stance from its maximum vertical displacement during swing. Additionally, the posture of the rats was evaluated by measuring the height of the iliac crest during the gait cycle and comparing it to intact rats. It's important to note that the ladder and open-field scoring, as well as kinematic analysis, were conducted offline.

## Awake motor maps

Motor maps were performed in awake animals, as in *Bonizzato and Martinez, 2021*. In contrast to traditional cortical mapping, which is performed under ketamine anesthesia and during terminal experiments, we implanted 32-channel electrode arrays chronically within the motor cortex of rats. We monitored changes in corticospinal transmission by recording hindlimb movements evoked by intra-cortical stimulation. Awake mapping offers two primary advantages: first, it enables the longitudinal tracking of motor cortex plasticity in the same animal, and second, it reveals non-pyramidal transmission, which is suppressed by ketamine anesthesia (*Bonizzato and Martinez, 2021*).

The 32-channel cortical implant was connected to a 32-channel stimulator (Tucker-Davis Technologies). A 40 ms pulse train (330 Hz, biphasic, 200 µs/phase) was delivered to each site, and hindlimb responses were visually assessed while the animal was at rest and supported by trunk support. We initiated testing with stimulation amplitudes of 100 µA, evaluating the response type (proximal or distal), and identifying the minimum amplitude that evoked a visible twitch as the threshold value. Testing was interrupted when no response was detected. A joint motor map was constructed using data from 12 subjects, selecting the most frequent response for each site across the population (*Figure 6*). In the case of two rats, wherein specific channels predominantly evoked ipsilateral motor responses, we recorded EMG signals during an additional 10 rounds of testing for all channels. Following the normalization of each muscle activity to spontaneous locomotion, we quantified the ipsilateral dominance of muscle activation as the ratio of the left and right tibialis anterior evoked responses (*Figure 7D*).

## Phase-coherent cortical stimulation

The phase-coherent neurostimulation strategy has been previously described in detail (*Bonizzato and Martinez, 2021*). During treadmill locomotion, EMG activity was processed online, and a trigger event was detected when the signal crossed a manually selected activation threshold. Subsequently, a biphasic 40 ms train at 330 Hz was delivered with a specific delay. Among the 32 sites of the cortical array, the stimulation channel that evoked the strongest right hindlimb flexion (or left hindlimb flexion in the case of ipsilateral modulation) during motor maps assessment was chosen.

For amplitude characterization, the left flexor served as the synchronization signal and the delay was fixed, corresponding to 140–190 ms depending on the rat's gait pattern. In this protocol, the stimulation was delivered in the late right stance or the corresponding early left stance. The amplitude values were linearly spaced within a functional range, defined from a minimum visible effect (40–100 µA before injury, 25–70 µA after injury) to a maximum comfortable value for the animal (125–300 µA before injury, 70–200 µA after injury). For each episode of locomotion, a single, fixed stimulation amplitude was selected from within the defined range, and all amplitude values within the range were systematically tested in subsequent episodes.

Regarding timing characterization, synchronization was alternatively based on the right flexor and the left flexor or right extensor activity. The amplitude was fixed and equal to a medium value of the functional range. The delay varied among trials to ensure that stimulation complementarily covered the entire gait cycle (0–200 ms for the flexors, 80–280 ms for the extensor, in steps of 40 ms).

In specific cases involving ipsilateral kinematics modulation, the trigger was detected from the right flexor signal, the delay was fixed (160 ms, corresponding to late left stance), and the amplitudes varied within the functional range (lower bound 50 and 100 µA, upper bound 200 and 175 µA). Random permutation of trials was employed whenever possible in each characterization.

## Long-train cortical stimulation

For each tested rat, we initially selected the cortical channel that elicited the strongest hindlimb responses through visual observation. A total of six awake resting rats were involved in the experiments. During these experiments, we provided manual support to the rats at the torso and forelimbs, allowing the hindlimbs to remain relaxed without any support. We recorded the hindlimb responses to long-train stimuli, which consisted of 250 ms duration, 330 Hz frequency, biphasic pulses with cathodic first phases, and a pulse width of 200 µs/phase. These responses were captured using a camera recording at 120 Hz. In three of the rats, we also collected EMG data from both ankle flexor muscles (tibialis anterior) concurrently, with a sampling rate of 6 kHz. The stimulus amplitude for all long-train experiments was fixed at 100 µA, following established protocols (*Brown and Teskey, 2014*; *Singleton et al., 2021*). These experiments were conducted in both the intact state and weekly for 3

weeks after SCI. In addition, we administered a single dose of ketamine (120 mg/kg, intraperitoneal [IP]) to four intact rats to confirm the absence of alternated evoked rhythms under ketamine-induced sedation. These rats were tested 10 min after the ketamine injection, during a moderately sedated state where corneal and paw withdrawal reflexes were preserved, but no overt spontaneous movement occurred. In this ketamine-administered condition, a stimulus amplitude of 150 μA was used.

## Current spread

To investigate the potential propagation of current to the homologous motor cortex, we conducted experiments involving the ablation of the contralateral motor cortex following SCI. We assessed the immediate effects of ipsilesional motor cortex stimulation on movement modulation and posture in three rats (*Figure 8*).

## Histology

At the conclusion of the experiments, euthanasia was performed on the rats using pentobarbital administration (Euthanyl, 100 mg/kg, IP). Transcardiac perfusion was carried out using a 0.2% phosphate-buffered saline solution, followed by a 4% paraformaldehyde (PFA) solution (pH 7.4). The spinal cords were then extracted and initially placed in a 4% PFA solution, followed by immersion in a 20% sucrose solution. To assess the extent of lesions, spinal sections around the T9 segment were sliced into 40 μm sections, and tissue damage was examined under a microscope. Lesion profiles at the epicenter level were reconstructed, and the extent of healthy and damaged tissue was quantified.

## Quantification and statistical analyses

All results are presented as the mean value ± the standard error of the mean (SEM). The statistical analyses were conducted as follows: First, we assessed whether each population could be considered normally distributed or not. In cases where a population was trivially non-normally distributed (e.g. low amounts of dragging saturate to zero), non-parametric tests were applied. For all other cases, we did not make an automatic presumption of normality. Additionally, a one-sample Kolmogorov-Smirnov test was performed to test for normality.

After determining the distribution, statistical tests were carried out between populations. For all analyses where replicates were individual rats and both populations' normality could not be excluded, we used the paired Student's t-test. In cases involving other populations or where normality could be excluded, the Wilcoxon signed-rank test was employed for paired population samples.

In analyses where replicates were individual gait cycles and both populations' normality could not be excluded, we used the unpaired Student's t-test. For other populations or cases where normality could be excluded, the Wilcoxon rank-sum test was utilized for non-paired tests.

The specific test used is always indicated alongside the p-value. All tests were one-sided, as our hypotheses were strictly defined to predict motor improvement. Specifically, we hypothesized that delivering an extension-inducing stimulus would enhance leg extension, and delivering a flexion-inducing stimulus would enhance leg flexion. Consequently, any potentially statistically significant result in the opposite direction (e.g. inhibition) would not be considered. However, no such occurrences were observed.

The study was powered for comparison between no stimulation and maximum stimulation only, which was the only statistical comparison performed in each figure. When replicates were rats, power analysis assumed effect sizes of 2.5 times the sample variability, requiring n=5 for a β=0.8 probability of obtaining significant effects (α<0.05). When replicates were single gait cycles, power analysis assumed effect sizes of 1.50 times the sample variability, necessitating n=7 for a β=0.8 probability of obtaining significant effects. However, 10 gait cycles were used to provide extra room for unexpected variability. Intermediate stimulation values are reported to demonstrate proportionality and were assessed with linear fits, with adequacy measured using VAF. Samples with p<α were considered statistically significant.

## Acknowledgements

The authors would like to thank Émilie Délage and Victorine Artot for their participation in data analysis; Andrew Brown and Mohamad Sawan for the fruitful discussion on this work's Materials and methods; Philippe Drapeau and Marc Bourdeau for technical assistance; Marjolaine Homier, Stéphane Ménard, Raphaël Santamaria, and the staff at the Division des Animaleries for supporting our animal care. This work was supported by the Craig H Neilsen Foundation and the Natural Sciences and Engineering Research Council of Canada. MM was supported by a salary award from Fonds de Recherche Québec-Santé (FRQ-S). MB was supported by fellowships from the FRQ-S, the Institut de valorisation des données (IVADO), the TransMedTech Institute, and a departmental postdoctoral fellowship in memory of Tomás A Reader. EM was supported by a fellowship from the TransMedTech Institute.

## Additional information

### Competing interests

Marco Bonizzato: MB submitted an international patent application (U.S. No. 62/880,364) covering a device allowing performing coherent cortical stimulation during locomotion. He is also shareholders of a start-up company focused on developing neurostimulation technologies, 12576830 Canada Inc. Marina Martinez: MM submitted an international patent application (U.S. No. 62/880,364) covering a device allowing performing coherent cortical stimulation during locomotion. She is also shareholders of a start-up company focused on developing neurostimulation technologies, 12576830 Canada Inc. The other authors declare that no competing interests exist.

### Funding

| Funder | Grant reference number | Author |
| --- | --- | --- |
| Craig H. Neilsen Foundation | | Marina Martinez |
| Natural Sciences and Engineering Research Council of Canada | | Marina Martinez |

The funders had no role in study design, data collection and interpretation, or the decision to submit the work for publication.

### Author contributions

Elena Massai, Data curation, Formal analysis, Investigation, Methodology, Writing – original draft, Writing – review and editing; Marco Bonizzato, Conceptualization, Data curation, Formal analysis, Investigation, Methodology, Writing – original draft, Writing – review and editing; Isley De Jesus, Roxanne Drainville, Formal analysis; Marina Martinez, Conceptualization, Resources, Data curation, Formal analysis, Supervision, Funding acquisition, Validation, Visualization, Methodology, Project administration, Writing – review and editing

### Author ORCIDs

Marina Martinez ⓘ https://orcid.org/0000-0002-3294-3017

### Ethics

All procedures adhered to the guidelines established by the Canadian Council of Animal Care and received approval from the Comité de déontologie de l'expérimentation sur les animaux (CDEA), the animal ethics committee at Université de Montréal (protocol #17-083).

Reviewer #2 (Public review): https://doi.org/10.7554/eLife.92940.3.sa1
Reviewer #3 (Public review): https://doi.org/10.7554/eLife.92940.3.sa2
Author response https://doi.org/10.7554/eLife.92940.3.sa3

## Additional files

### Supplementary files

• Supplementary file 1. List of animals engaged in each experiment. Rats marked with * did not receive left motor cortex implantation. They were included in the study for establishing spontaneous changes in posture over time (*Figure 5A and B*).

• MDAR checklist

### Data availability

All data generated or analysed during this study are included in the supporting files.

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
