## [Editor Report · eLife Assessment]

The contributions of ipsilateral cortical pathways to motor control are yet not fully understood. Here, the authors present **important** insights into their role in locomotion following unilateral spinal cord injury. Their data provide **convincing** evidence in rats that stimulation of ipsilateral motor cortex improves the injured side's ability to support weight and leads to improved locomotion, a result that may inspire new treatments for spinal or cerebral injuries.

---

## [Referee Report · Reviewer #2 (Public review)]

Summary:

The authors long term goals are to understand the utility of precisely phased cortex stimulation regimes on recovery of function after spinal cord injury (SCI). In prior work the authors explored effects of contralesion cortex stimulation. Here, they explore ipsilesion cortex stimulation in which the ipsilesion corticospinal fibers that cross at the pyramidal decussation are spared. The authors explore the effects of such stimulation in intact rats and rats with a hemisection lesion at thoracic level ipsilateral to the stimulated cortex. The appropriately phased microstimulation enhances contralateral flexion and ipsilateral extension, presumably through lumbar spinal cord crossed extension interneuron systems. This microstimulation improves weight bearing in the ipsilesion hindlimb soon after injury, before any normal recovery of function would be seen. The contralateral homologous cortex can be lesioned in intact rats without impacting the microstimulation effect on flexion and extension during gait. In two rats ipsilateral flexion responses are noted, but these are not clearly demonstrated to be independent of the contralateral homologous cortex remaining intact.

Strengths:

This paper adds to prior data on cortical microstimulation by the authors' laboratory in interesting ways. First, the strong effects of the spared crossed fibers from ipsi-lesional cortex in parts of the ipsi-lesion leg's step cycle and weight support function are solidly demonstrated. This raises the interesting possibility that stimulating contra-lesion cortex as reported previously may execute some of its effects through callosal coordination with the ipsi-lesion cortex tested here. This is also now discussed by the authors and may represent a significant aspect of these data. The authors demonstrate solidly that ablation of the contra-lesional cortex does not impede the effects reported here. I believe this has not been shown for the contra-lesional cortex microstimulation effects reported earlier, but I may be wrong.

Effects and neuroprosthetic control of these effects are explored well in the ipsi-lesion cortex tests here.

Weaknesses:

Some data is based on only a few rats. For example (N=2) for ipsilateral flexion effects of microstimulation. N=3 for homologous cortex ablation, and only ipsi extension is tested it seems. However, these data clearly point the way and replication is likely.

Likely Impacts:

This data adds in significant ways to prior work by the authors, and an understanding of how phased stimulation in cortical neuroprosthetics may aid in recovery of function after SCI, especially if a few ambiguities in writing and interpretation are fully resolved.

---

## [Referee Report · Reviewer #3 (Public review)]

Summary:

This article aims to investigate the impact of neuroprosthesis (intracortical microstimulation) implanted unilaterally on the lesion side in the context of locomotor recovery following thoracic spinal hemisection.

Strength:

The study reveals that stimulating the left motor cortex, on the same side as the lesion, not only activates the expected right (contralateral) muscle activity but also influences unexpected muscle activity on the left (ipsilateral) side. These muscle activities resulted a substantial enhancement in lift during the swing phase of the contralateral limb and improved trunk-limb support for the ipsilateral limb. They used different experimental and stimulation condition to show the ipsilateral limb control evoked by the stimulation. This outcome holds significance, shedding light on the engagement of the contralateral-projecting corticospinal tract (CST) in activating a not only contralateral but also ipsilateral spinal network.

The experimental design and findings align with the investigation of the stimulation effect of contralateral projecting CSTs. They carefully examined the recovery of ipsilateral limb control with motor maps. And they also tested the effective sites of cortical stimulation. The study successfully demonstrates the impact of electrical stimulation on the contralateral projecting neurons on ipsilateral limb control during locomotion, as well as identifying importance stimulation spots for such effect. These results contribute to our understanding of how these neurons influence bilateral spinal circuitry. The study's findings contribute valuable insights to the broader neuroscience and rehabilitation communities.

Weakness:

The term "ipsilateral" lacks a clear definition in some cases, potentially causing confusion for the reader. Readers can potentially link ipsilateral cortical network to ipsilateral-projecting CSTs, which is less likely to play a role to ipsilateral limb control in this study since this tract is disrupted by the thoracic hemisection.

Specific comments:

Abstract: Line 1-4: Consider refining the initial sentences of the abstract to reduce ambiguity around the term 'ipsilateral lesion' and its potential conflation with ipsilateral projecting cortical neurons.

The abstract begins with 'Control of voluntary limb movement is predominantly attributed to the contralateral motor cortex.' This is followed by, 'However, increasing evidence suggests the involvement of ipsilateral cortical networks in this process, especially in motor tasks requiring bilateral coordination, such as locomotion.'

The phrase 'ipsilateral cortical networks' remains somewhat unclear. Readers may mistakenly interpret it as referring to the ipsilateral projecting corticospinal tract (CST), which is not the focus of this study.

Shifting the focus away from 'ipsilateral cortical control' and instead highlighting ipsilateral limb control following a spinal hemisection would improve clarity. This adjustment would also align the title and abstract more closely with the study's primary focus.

Introduction:

It is suggested to revise the introduction to more closely align with the study's experimental design and outcomes, placing emphasis on the stimulation effects observed in contralateral projecting tracts rather than implying a primary focus on ipsilateral projecting CST neurons.

Line 30-32: "Nevertheless, the function of the ipsilateral motor cortex is unclear and its role in the recovery of motor control after injury remains controversial. " This still gives the impression that ipsilateral projecting CST is the topic of the research here. Also, some of the cited references contains discuss ipsilateral projecting CSTs.

Line 34-36: "While the most prominent feature of motor cortex pathways is their contralateral organization, unilateral or bilateral movements are well represented in the ipsilateral hemisphere." This sentence is unclear to me. It would be helpful to specify what 'ipsilateral hemisphere' refers to-ipsilateral to what? Clarifying whether it's ipsilateral to the lesion or another reference point would make the statement more precise."

---

## [Author Response]

The following is the authors’ response to the original reviews.

**eLife Assessment**
This manuscript reveals important insights into the role of ipsilateral descending pathways in locomotion, especially following unilateral spinal cord injury. The study provides solid evidence that this method improves the injured side's ability to support weight, and as such the findings may lead to new treatments for stroke, spinal cord injuries, or unilateral cerebral injuries. However, the methods and results need to be better detailed, and some of the statistical analysis enhanced.

Thank you for your assessment. We incorporated various text improvements in the final version of the manuscript to address the weaknesses you have pointed out. The specific improvements are outlined below.

**Public Reviews:**

**Reviewer #1 (Public Review):**
Summary:This manuscript provides potentially important new information about ipsilateral cortical impact on locomotion. A number of issues need to be addressed.Strengths:The primary appeal and contribution of this manuscript are that it provides a range of different measures of ipsilateral cortical impact on locomotion in the setting of impaired contralateral control. While the pathways and mechanisms underlying these various measures are not fully defined and their functional impacts remain uncertain, they comprise a rich body of results that can inform and guide future efforts to understand cortical control of locomotion and to develop more effective rehabilitation protocols.Weaknesses:(1) The authors state that they used a cortical stimulation location that produced the largest ankle flexion response (lines 102-104). Did other stimulation locations always produce similar, but smaller responses (aside from the two rats that showed ipsilateral neuromodulation)? Was there any site-specific difference in response to stimulation location?

We derived motor maps in each rat, akin to the representation depicted in Fig 6. In each rat, alternative cortical sites did, indeed, produce distal or proximal contralateral leg flexion responses. Distal responses were more likely to be evoked in the rostral portion of the array, similarly to proximal responses early after injury. This distribution in responses across different cortical sites is reported in this study (Fig. 6) and is consistent with our prior work. The Results section has been revised to provide additional clarification of the passage you indicated and context for the data presented in Figure 6:

On page 4, we have clarified: “Stimulation through these channels produced a strong whole-leg flexion movement, with an evident distal component. From visual inspection, all responding electrodes in the array produced contralateral leg flexion, although with different strength of contraction for a fixed stimulation intensity (100μA). Moreover, some sites did not present a distal movement component, failing in eliciting ankle flexion and resulting in a generally weaker proximal flexion.”

On page 12, we have further noted: “By visually inspecting the responses elicited by stimulation delivered through each of the array electrodes, we categorized movements as proximal or distal. This classification was based on whether the ankle participated in the evoked response or if the movement was restricted to the proximal hindlimb. Each leg was scored independently.”

(2) Figure 2: There does not appear to be a strong relationship between the percentage of spared tissue and the ladder score. For example, the animal with the mild injury (based on its ladder score) in the lower left corner of Figure 2A has less than 50% spared tissue, which is less spared tissue than in any animal other than the two severe injuries with the most tissue loss. Is it possible that the ladder test does not capture the deficits produced by this spinal cord injury? Have the authors looked for a region of the spinal cord that correlates better with the deficits that the ladder test produces? The extent of damage to the region at the base of the dorsal column containing the corticospinal tract would be an appropriate target area to quantify and compare with functional measures.

In Fig. S6 of our 2021 publication "Bonizzato and Martinez, Science Translational Medicine", we investigated the predictive value of tissue sparing in specific sub-regions of the spinal cord for ladder performance. Among others, we examined the correlation between the accuracy of left leg ladder performance in the acute state and the preservation of the corticospinal tract (CST). Our results indicated that dorsal CST sparing serves as a mild predictor for ladder deficits, confirming the results obtained in this study.

(3) Lines 219-221: The authors state that "phase-coherent stimulation reinstated the function of this muscle, leading to increased burst duration (90{plus minus}18% of the deficit, p=0.004, t-test, Fig. 4B) and total activation (56{plus minus}13% of the deficit, p=0.014, t-test, Fig. 3B). This way of expressing the data is unclear. For example, the previous sentence states that after SCI, burst duration decreased by 72%. Does this mean that the burst duration after stimulation was 90% higher than the -72% level seen with SCI alone, i.e., 90% + -72% = +18%? Or does it mean that the stimulation recovered 90% of the portion of the burst duration that had been lost after SCI, i.e., -72% * (100%-90%) = -7%? The data in Figure 4 suggests the latter. It would be clearer to express both these SCI alone and SCI plus stimulation results in the text as a percent of the pre-SCI results, as done in Figure 4.

Your assessment is correct; we intended to report that the stimulation recovered 90% of the portion of the burst duration that had been lost after SCI. This point has been clarified (see page 9):

“…leading to increased burst duration (recovered 90±18% of the lost burst duration, p=0.004, t-test, Fig. 4B) and total activation (recovered 56±13% of the total activation, p=0.014, t-test, Fig. 3B)”

(4) Lines 227-229: The authors claim that the phase-dependent stimulation effects in SCI rats are immediate, but they don't say how long it takes for these effects to be expressed. Are these effects evident in the response to the first stimulus train, or does it take seconds or minutes for the effects to be expressed? After the initial expression of these effects, are there any gradual changes in the responses over time, e.g., habituation or potentiation?

The effects are immediately expressed at the very first occurrence of stimulation. We never tested a rat completely naïve to stimuli, as each treadmill session involves prior cortical mapping to identify a suitable active site for involvement in locomotor experiments. Yet, as demonstrated in Supplementary Video 1 accompanying our 2021 publication on contralateral effects of cortical stimulation, "Bonizzato and Martinez, Science Translational Medicine," the impact of phase-dependent cortical stimulation on movement modulation is instantaneous and ceases promptly upon discontinuation of the stimulation. We did not quantify potential gradual changes in responsiveness over time, but we cannot exclude that for long stimulation sessions (e.g., 30 min or more), stimulus amplitude may need to be slightly increased over time to compensate habituation.

(5) Awake motor maps (lines 250-277): The analysis of the motor maps appears to be based on measurements of the percentage of channels in which a response can be detected. This analytic approach seems incomplete in that it only assesses the spatial aspect of the cortical drive to the musculature. One channel could have a just-above-threshold response, while another could have a large response; in either case, the two channels would be treated as the same positive result. An additional analysis that takes response intensity into account would add further insight into the data, and might even correlate with the measures of functional recovery. Also, a single stimulation intensity was used; the results may have been different at different stimulus intensities.

We confirm that maps of cortical stimulation responsiveness may vary at different stimulus amplitudes. To establish an objective metric of excitability, we identified 100µA as a reliable stimulation amplitude across rats and used this value to build the ipsilateral motor representation results in Figure 6. This choice allows direct comparison with Figure 6 of our 2021 article, related to contralateral motor representation. The comparison reveals a lack of correlation with functional recovery metrics in the ipsilateral case, in contrast to the successful correlation achieved in the contralateral case.

Regarding the incorporation of stimulation amplitudes into the analysis, as detailed in the Method section (lines 770-771), we systematically tested various stimulation amplitudes to determine the minimal threshold required for eliciting a muscle twitch, identified as the threshold value. This process was conducted for each electrode site.

Upon reviewing these data, we considered the possibility of presenting an additional assessment of ipsilateral cortical motor representation based on stimulation thresholds. However, the representation depicted in the figure did not differ significantly from the data presented in Figure 6A. Furthermore, this representation introduced an additional weakness, as it was unclear how to represent the absence of a response in the threshold scale. We chose to arbitrarily designate it as zero on the inverse logarithmic scale, where, for reference, 100 µA is positioned at 0.2 and 50 µA at 0.5.

In conclusion, we believe that the conclusions drawn from this analysis align substantially with those in the text. The addition of the threshold analysis, in our assessment, would not contribute significantly to improving the manuscript.

**Author response image 1. sa3fig1:** Threshold analysis.

**Author response image 2. sa3fig2:** Occurrence probability analysis, for comparison.

(6) Lines 858-860: The authors state that "All tests were one-sided because all hypotheses were strictly defined in the direction of motor improvement." By using the one-sided test, the authors are using a lower standard for assessing statistical significance that the overwhelming majority of studies in this field use. More importantly, ipsilateral stimulation of particular kinds or particular sites might conceivably impair function, and that is ignored if the analysis is confined to detecting improvement. Thus, a two-sided analysis or comparable method should be used. This appropriate change would not greatly modify the authors' current conclusions about improvements.

Our original hypothesis, drawn from previous studies involving cortical stimulation in rats and cats, as well as other neurostimulation research for movement restoration, posited a favorable impact of neurostimulation on movement. Consistent with this hypothesis, we designed our experiments with a focus on enhancing movement, emphasizing a strict direction of improvement.

It's important to note that a one-sided test is the appropriate match for a one-sided hypothesis, and it is not a lower standard in statistics. Each experiment we conducted was constructed around a strictly one-sided hypothesis: the inclusion of an extensor-inducing stimulus would enhance extension, and the inclusion of a flexion-inducing stimulus would enhance flexion. This rationale guided our choice of the appropriate statistical test.

We acknowledge your concern regarding the potential for ipsilateral stimulation to have negative effects on locomotion, which might not be captured when designing experiments based on one-sided hypotheses. That is, when hypothesizing that an extensor stimulus would enhance extension (a one-sided hypothesis) in a functional task, and finding an opposite result (inhibition), statistical rigor would impose that we cannot present that result as significant. This concern is valid, and we explicitly mentioned our design choice it in the method section, Quantification and statistical analyses:

“All tests were one-sided, as our hypotheses were strictly defined to predict motor improvement. Specifically, we hypothesized that delivering an extension-inducing stimulus would enhance leg extension, and delivering a flexion-inducing stimulus would enhance leg flexion. Consequently, any potentially statistically significant result in the opposite direction (e.g., inhibition) would not be considered. However, no such occurrences were observed.”

As a final note, even if such opposite observations were made, they could serve as the basis for triggering an ad-hoc follow-up study.

Reviewer #1 also provided several detailed suggestions in the section “Recommendations for the authors”. We estimated that each of them was beneficial for the correctness or for the readability of the text, and thus all were incorporated into the final version.

**Reviewer #2 (Public Review):**
Summary:The authors' long-term goals are to understand the utility of precisely phased cortex stimulation regimes on recovery of function after spinal cord injury (SCI). In prior work, the authors explored the effects of contralesion cortex stimulation. Here, they explore ipsilesion cortex stimulation in which the corticospinal fibers that cross at the pyramidal decussation are spared. The authors explore the effects of such stimulation in intact rats and rats with a hemisection lesion at the thoracic level ipsilateral to the stimulated cortex. The appropriately phased microstimulation enhances contralateral flexion and ipsilateral extension, presumably through lumbar spinal cord crossed-extension interneuron systems. This microstimulation improves weight bearing in the ipsilesion hindlimb soon after injury, before any normal recovery of function would be seen. The contralateral homologous cortex can be lesioned in intact rats without impacting the microstimulation effect on flexion and extension during gait. In two rats ipsilateral flexion responses are noted, but these are not clearly demonstrated to be independent of the contralateral homologous cortex remaining intact.Strengths:This paper adds to prior data on cortical microstimulation by the laboratory in interesting ways. First, the strong effects of the spared crossed fibers from the ipsi-lesional cortex in parts of the ipsi-lesion leg's step cycle and weight support function are solidly demonstrated. This raises the interesting possibility that stimulating the contra-lesion cortex as reported previously may execute some of its effects through callosal coordination with the ipsi-lesion cortex tested here. This is not fully discussed by the authors but may represent a significant aspect of these data. The authors demonstrate solidly that ablation of the contra-lesional cortex does not impede the effects reported here. I believe this has not been shown for the contra-lesional cortex microstimulation effects reported earlier, but I may be wrong. Effects and neuroprosthetic control of these effects are explored well in the ipsi-lesion cortex tests here.

In the revised version of the manuscript, we incorporated various text improvements to address the points you have highlighted in your review. Additionally, we have integrated the suggested discussion topic on callosal coordination related to contralateral cortical stimulation. The discussion section now incorporates:

“Since bi-cortical interactions in sculpting descending commands are known (Brus-Ramer et al., 2009), and in light of the changes we report in ipsilesional motor cortex excitability, the role of the ipsilateral cortex in mediating or supporting functional descending commands from the contralateral cortex, particularly the immediate increase in flexion of the affected hindlimb and long-term recovery of functional control (Bonizzato & Martinez, 2021), could be further explored.”

The localization of the specific channels closest to the interhemispheric fissure (Fig. 7D) may suggest the involvement of transcallosal interactions in mediating the transmission of the cortical command generated in the ipsilateral motor cortex (Brus-Ramer, Carmel, & Martin, 2009). “While ablation experiments (Fig. 8) refute this hypothesis for ipsilateral extension control, they do not conclusively determine whether a different efferent pathway is involved in ipsilateral flexion control in this specific case."

Weaknesses:Some data is based on very few rats. For example (N=2) for ipsilateral flexion effects of microstimulation. N=3 for homologous cortex ablation, and only ipsi extension is tested it seems. There is no explicit demonstration that the ipsilateral flexion effects in only 2 rats reported can survive the contra-lateral cortex ablation.

We agree with this assessment. The ipsilateral flexion representation is here reported as a rare but consistent phenomenon, which we believe to have robustly described with Figure 7 experiments. We underlined in the text that the ablation experiment did not conclude on the unilateral-cortical nature of ipsilateral flexion effects, by replacing the sentence with the following:

“While ablation experiments (Fig. 8) refute this hypothesis for ipsilateral extension control, they do not conclusively determine whether a different efferent pathway is involved in ipsilateral flexion control in this specific case."

Some improvements in clarity and precision of descriptions are needed, as well as fuller definitions of terms and algorithms.Likely Impacts: This data adds in significant ways to prior work by the authors, and an understanding of how phased stimulation in cortical neuroprosthetics may aid in recovery of function after SCI, especially if a few ambiguities in writing and interpretation are fully resolved.

The manuscript text has been revised in its final version, and we sought to eliminate all ambiguity in writing and data interpretation.

In the section “Recommendations for the authors” Reviewer #2 also suggested to better define multiple terms throughout the manuscript. A clarification was added for each.

The Reviewer pointed out that we might have overlooked a correlation between locomotor recovery and motor maps increase in Figure 6. We re-approached this evaluation and found that the reviewer is correct. We were led to think that there was no correlation by “horizontally” looking at whether motor map size across rats would predict locomotor scores (as it did in the case of contralateral cortex mapping, Bonizzato and Martinez, 2021). However we now found a strong correlation between changes that happen over time for each rat and locomotor recovery, a result that was only hinted with no appropriate quantification in the previous version of the manuscript. We have now reformulated the results of Figure 6 on page 12, to include this result, and we would like to thank the reviewer for having noticed this opportunity.

Finally, we have expanded the discussion to include the following points:

The possibility that hemi-cortex coordination of contralesional microstimulation inputs may explain the Sci Transl Med results for contralesional cortex ICMS, which warrants further investigation.

The recognition that the ablation experiments do not provide conclusive evidence regarding ipsilateral flexion control and whether an alternative efferent pathway might be involved in this specific case.

**Reviewer #3 (Public Review):**
Summary:This article aims to investigate the impact of neuroprosthesis (intracortical microstimulation) implanted unilaterally on the lesion side in the context of locomotor recovery following unilateral thoracic spinal cord injury.Strength:The study reveals that stimulating the left motor cortex, on the same side as the lesion, not only activates the expected right (contralateral) muscle activity but also influences unexpected muscle activity on the left (ipsilateral) side. These muscle activities resulted in a substantial enhancement in lift during the swing phase of the contralateral limb and improved trunk-limb support for the ipsilateral limb. They used different experimental and stimulation conditions to show the ipsilateral limb control evoked by the stimulation. This outcome holds significance, shedding light on the engagement of the "contralateral projecting" corticospinal tract in activating not only the contralateral but also the ipsilateral spinal network.The experimental design and findings align with the investigation of the stimulation effect of contralateral projecting corticospinal tracts. They carefully examined the recovery of ipsilateral limb control with motor maps. They also tested the effective sites of cortical stimulation. The study successfully demonstrates the impact of electrical stimulation on the contralateral projecting neurons on ipsilateral limb control during locomotion, as well as identifying important stimulation spots for such an effect. These results contribute to our understanding of how these neurons influence bilateral spinal circuitry. The study's findings contribute valuable insights to the broader neuroscience and rehabilitation communities.

Thank you for your assessment of this manuscript. The final version of the manuscript incoporates your suggestions for improving term clarity and we enhanced the discussion on the mechanisms of spinal network engagement, as outlined below.

Weakness:The term "ipsilateral" lacks a clear definition in the title, abstract, introduction, and discussion, potentially causing confusion for the reader.[and later] However, in my opinion, readers can easily link the ipsilateral cortical network to the ipsilateral-projecting corticospinal tract, which is less likely to play a role in ipsilateral limb control in this study since this tract is disrupted by the thoracic spinal injury.

In order to mitigate the risk of having readers linking the effects of ipsilateral cortical stimulation with ipsilateral-projecting corticospinal tract, we specified:

In the abstract, we precise that our goal was: “to investigate the functional role of the ipsilateral motor cortex in rat movement through spared contralesional pathways.”

In the introduction: “In most cases, this lesion also disrupts all spinal tracts descending on the same side as the cortex under investigation at the thoracic level, meaning that the transmission of cortical commands to the ipsilesional hindlimb must depend on crossed descending tracts (Fig. S1).”

The unexpected ipsilateral (left) muscle activity is most likely due to the left corticospinal neurons recruiting not only the right spinal network but also the left spinal network. This is probably due to the joint efforts of the neuroprosthesis and activation of spinal motor networks which work bilaterally at the spinal level.

We agree with your assessment and the discussion section now emphasizes the effects of supraspinal drive onto spinal circuits.

In the section “Recommendations for the authors” Reviewer #3 suggested to provide an early reminder to the reader that the focus is on exploring the control of the ipsilateral limb through the corticospinal tract of the same side, projecting contralaterally. We did so in the abstract and introduction, as presented above.

The reviewer also suggested that the discussion could be shorter. While we recognize it covers diverse subjects that may appeal to different readers, we believe omitting some sections could limit its overall scope. The manuscript underwent three revisions and a thorough dialogue with reviewers from diverse backgrounds, and we are hesitant to undo some of these improvements.

Moreover, the section falls short of fully exploring the involvement of contralateral projecting corticospinal neurons in spinal networks for diverse motor behaviors. It could potentially delve into aspects like the potential impact of corticospinal inputs on gating the cross-extensor reflex loop and elucidating the mechanisms underlying the recruitment of the ipsilateral spinal network for generating ipsilateral limb movements. Is it a direct control on motor neurons or via existing spinal circuits?

The discussion section now includes the potential spinal circuits through which corticospinal neurons may affect motor control and reflexes.

Reviewer #3 also provided several detailed suggestions in the sub-section “Minor points”. We estimated that all of them were beneficial for the correctness or for the readability of the text, and thus were incorporated into the final version. Some of the questions raised were answered directly in the text (defining “% of chronic map” and rephrasing the original Line 479). We would like to answer here below two remaining questions:

Fig. 3C I wonder what is the average latency between stimulation onset and onset of right ankle flexor activity. Is the latency fixed, or variable (which probably indicates that the Cortical activation signal is integrated with spinal CPG activity.)

ICMS trains, unfortunately, do not allow for precise dissection of transmission timing. Single pulses at 100 µA are insufficient to generate motoneuron responses and require multiple pulses to build up cortical transmission. Alstermark et al. (Journal of Neurophysiology, 2004) used two to four stimuli with higher amplitudes to investigate forelimb transmission timing. In our 2021 Science Translational Medicine paper, we employed single pulses at 1 mA to establish transmission delays from the contralateral cortex to the ankle flexor. However, the circuits recruited at 1 mA are not directly comparable to those activated by shorter trains.

In this study, we used cortical trains of approximately 14 pulses, typical of ICMS protocols. Each pulse could potentially be the first to generate a response volley in the ankle flexor, with delays measured at 30 to 60 ms from ICMS train onset. While we believe that cortical commands are necessarily integrated with spinal CPG activity—as indicated in Figures 1B and 3D, where timing is crucial and descending commands can be gated out if delivered off-phase—the variability in latency that we recorded could be attributed to any of the following factors: cortical activation build-up, integration within reticular relay networks, or CPG integration.

Fig. 4A. Why is the activity of under contralateral ankle flexor intact condition is later than the stimulation condition?

We timed the stimulation to coincide with the contralateral leg lift and did not adjust its onset relative to spontaneous walking in SCI rats. Although stimulation could induce leg lift, as shown in Fig. 4A, SCI rats exhibited a slightly earlier and stronger activation of the right (contralateral) ankle flexor muscle even during spontaneous walking. This phenomenon is attributed to the deficits observed on the left side. The stronger right leg bears the body weight, as illustrated in Fig. 3, and thus, during body advancement, the right leg is engaged sooner and more rapidly (with a shorter swing phase) to provide support (right foot forward).